

# Signatures of a critical point
# in the many-body localization transition

**Ángel L. Corps[1]⋆, Rafael A. Molina[2]† and Armando Relaño[1]‡**

**1** Departamento de Estructura de la Materia, Física Térmica y Electrónica & GISC,
Universidad Complutense de Madrid, Av. Complutense s/n, E-28040 Madrid, Spain
**2** Instituto de Estructura de la Materia, IEM-CSIC, Serrano 123, E-28006 Madrid, Spain

⋆ angelo04@ucm.es, † rafael.molina@csic.es, ‡ armando.relano@fis.ucm.es

## Abstract

Disordered interacting spin chains that undergo a many-body localization transition are characterized by two limiting behaviors where the dynamics are chaotic and integrable. However, the transition region between them is not fully understood yet. We propose here a possible finite-size precursor of a critical point that shows a typical finite-size scaling and distinguishes between two different dynamical phases. The *kurtosis excess of the diagonal fluctuations* of the full one-dimensional momentum distribution from its microcanonical average is maximum at this singular point in the paradigmatic disordered $J_1$-$J_2$ model. For system sizes accessible to exact diagonalization, both the position and the size of this maximum scale linearly with the system size. Furthermore, we show that this singular point is found at the same disorder strength at which the Thouless and the Heisenberg energies coincide. Below this point, the spectral statistics follow the universal random matrix behavior up to the Thouless energy. Above it, no traces of chaotic behavior remain, and the spectral statistics are well described by a generalized semi-Poissonian model, eventually leading to the integrable Poissonian behavior. We provide, thus, an integrated scenario for the many-body localization transition, conjecturing that the critical point in the thermodynamic limit, if it exists, should be given by this value of disorder strength.



# 1   Introduction

In the classical world the connection between the ergodic properties of a system and its class of regularity (chaotic and integrable) is well understood both mathematically and phenomenologically [1, 2]. The classical phase space is completely covered by the erratic trajectories followed by classical chaotic systems while integrability means that only a certain subset defined by the constraints of the constants of motion is accessible. In the first case the system will thermalize after being driven away from thermal equilibrium while in the second case it will generally not. Thus ergodicity and chaotic behavior are generally tied together and one is expected if the other is present.

In the quantum realm, however, the situation is much more involved. Our understanding of thermalization lies in the eigenstate thermalization hypothesis (ETH) [3–8]. According to this theory, a quantum system will thermalize if the fluctuations of the long-time averages of local observables around their standard microcanonical equilibrium value decrease fast enough with system size. Plainly put, the ETH states that the diagonal matrix elements of observables in the eigenstates of the system Hamiltonian must change with energy smoothly enough, coinciding with the microcanonical average up to some random and sufficiently small fluctuation. The connection between quantum thermalization and quantum chaotic behavior [9] is still an open question. The conjecture put forward by Bohigas, Giannoni and Schmit [10] indicates that the spectral fluctuations of quantum systems with an ergodic classical analogue follow exactly the predictions of random matrix theory (RMT) [11]. Hence the statistical analysis of the eigenlevel distribution of quantum spectra has been established as a very valuable tool to investigate quantum ergodicity. The ETH has been long thought to be a consequence of RMT, and thus the thermal properties of quantum systems seem to depend on the onset of quantum chaos in a non-trivial way. As in the classical case, quantum thermalization and chaos are two terms almost invariably associated with each other. However, the limits of applicability of RMT are more rigid than those of the ETH, and thus the latter can still apply beyond the capabilities of the former.

Originally, the RMT provided a solid framework to identify chaos in isolated quantum systems. Perhaps unexpectedly, the last decade has seen an enormous revival of the interest in spectral analysis due to the very exotic nature of quantum many-body systems. In this direction, disordered lattice models from condensed matter have gifted us with an unprecedented playground to test quantum ergodicity [12–14]. It seems by now well established that this is a property that clean (i.e., not disordered) quantum many-body systems generally have [4]. Introducing sufficiently strong disorder gives rise to one of the most striking exceptions to quantum thermal behavior: many-body localization (MBL) [3, 15], which is now taken as the prototypical nonergodic phenomenon. MBL is observed in some disordered lattice models

with local interactions such as the Heinsenberg spin-1/2 chain or its generalization to next to nearest-neighbor interactions, the $J_1$-$J_2$ model, for high enough disorder [16–28]. Systems in this class often display two completely opposed regimes: an ergodic phase for small disorder strength, in which both the ETH and RMT provide conclusive results, and the MBL phase, where the spectrum does not follow RMT and the system does not equilibrate at all. The region in between these two phases reveals highly unusual properties that have attracted very active investigation [29–31,33–35]. One of the most pressing questions arguably concerns the nature of the transition between the metallic, thermal phase and the insulating, nonthermal one, i.e., whether there is an actual phase transition in which a critical point can be observed or if we are dealing with a smooth crossover instead. Although exploring finite systems rather clearly leads to the second option, the answer in the thermodynamic limit (i.e., at macroscopic scales) is a lot more difficult to obtain, mainly due to the exponential increase of the dimensionality of the Hilbert space with system size in many-body quantum systems. A variety of ergodicity breaking indicators [15, 23, 24, 28, 36–43] have been used in the past to try and identify a hypothetical value of disorder strength that, in the thermodynamic limit, completely separates the ergodic and the nonergodic phases. The search for such a critical point implicitly assumes a nature of real phase transition in MBL systems, although this has not been proved and some apparent contradictions seem to exist which have led to much debate in the community [28, 44–47]. To obtain this value, different kinds of involved finite size scalings of such indicators have been employed. However, the problem is not simple at all and a strong dependence on the particular indicator and the number of sites means that some of the previously obtained values for the critical point are often incompatible with each other; today much uncertainty remains about the properties and the phenomenology of the MBL transition [19, 23, 28, 30, 38, 39, 44, 45, 47–51].

The aim of this paper is to introduce a new finite-size precursor of the critical point separating the transition between the ergodic phase and the many-body localized phase in disordered many-body quantum systems. The key quantity is the probability of extreme events in the fluctuations from the equilibrium microcanonical value of the diagonal matrix elements of the momentum distribution. These fluctuations are the basic quantities in the ETH. The probability of extreme events is quantified by the kurtosis excess of their ensemble probability distribution. The kurtosis excess is shown to have a maximum at the transition between the ergodic phase and the MBL phase in the $J_1$-$J_2$ disordered spin chain. Both the position and the value of this maximum are shown to increase linearly with the system size, at least for systems small enough to be exactly diagonalized. This finite-size scaling is compatible with recent results suggesting that the MBL transition belongs to the Berezinskii-Kosterlitz-Thouless class [47]. We also show that the maximum of the kurtosis excess happens at the same disorder strength at which spectral statistics cease to follow the universal random matrix theory behavior at any scale —when Thouless and Heisenberg energies coincide. In particular, we show that the disorder strength above which spectral fluctuations are well described by semi-poisson statistics shows the same finite-size scaling that the maximum of the kurtosis excess. Therefore, our results provide an integrated scenario for the MBL transition, involving both spectral statistics and thermalization.

The rest of this paper is structured as follows. In Section 2 we introduce the $J_1$-$J_2$ spin chain used in the computations. In Section 3, we present the concepts of quantum thermalization and ETH in subsection 3.1 which is at the core of our research. In the subsection 3.2 we introduce the diagonal fluctuations of the momentum distribution from their microcanonical averages and compute the results as a function of disorder, which is the main result of our paper. In subsection 3.3 we study the finite-size scaling of the proposed singular point. In Section 4 we study the spectral statistics across the transition introducing a full description including long-range correlations. In the first subsection 4.1 we present the semi-Poisson spec-

tral statistics that is followed by chains in the MBL phase; then, in the second subsection 4.2 we complement the previous results with the study of the Thouless energy $E_{\text{Th}}$ and long-range correlations on the ergodic side of the transition; finally, in subsection 4.3 we provide a general landscape of the MBL and ergodic phases in terms of their spectral properties. In Section 5 we relate the results of the spectral statistics with the behavior of the diagonal fluctuations presenting a coherent interpretation of the overall landscape, which suggest that looking at the value of disorder at which the Thouless time roughly equals the Heisenberg time emerges as a powerful criterion to elucidate the characteristics of MBL transitions. We gather the main conclusions of our work in Section 6.

## 2 Model: the disordered $J_1$-$J_2$ chain and many-body localization

Much of the research on the phenomenon of many-body localization has been carried out with the Heisenberg XXZ spin chain [16–28,52], which takes into account nearest-neighbors interactions only. In comparison, its simplest generalization, the $J_1$-$J_2$ model, has been somewhat less explored. However, it can be argued that the latter is a more generic model as it does not present Bethe ansatz integrability for the clean (i.e., disorder-free) case [53].

The $J_1$-$J_2$ model consists on a one-dimensional chain with $L$ sites, on-site magnetic fields $\omega_\ell$, and coupling parameters $J_1$ and $J_2$ where next to nearest-neighbors interactions are additionally considered. Its Hamiltonian can be cast in the form

$$
\begin{aligned}
\mathcal{H}(J_1, J_2) := \sum_{\ell=1}^{L} \omega_\ell \hat{S}_\ell^z + J_1 \sum_{\ell=1}^{L} \left( \hat{S}_\ell^x \hat{S}_{\ell+1}^x + \hat{S}_\ell^y \hat{S}_{\ell+1}^y + \lambda_1 \hat{S}_\ell^z \hat{S}_{\ell+1}^z \right) \\
+ J_2 \sum_{\ell=1}^{L} \left( \hat{S}_\ell^x \hat{S}_{\ell+2}^x + \hat{S}_\ell^y \hat{S}_{\ell+2}^y + \lambda_2 \hat{S}_\ell^z \hat{S}_{\ell+2}^z \right),
\end{aligned}
\tag{1}
$$

where $\hat{S}_\ell^{x,y,z}$ are the total spin-1/2 operators at site $\ell \in \{1, \ldots, L\}$. The Hamiltonian is simulated on a lattice with periodic boundary conditions as these minimize finite-size effects; thus, $\hat{S}_\ell^{x,y,z} = \hat{S}_{\ell+L}^{x,y,z}$. Disorder enters the chain via the uniformly, independently and randomly distributed magnetic fields $\omega_\ell \in [-\omega, \omega]$. The coupling constants $J_1$ and $J_2$ are associated to nearest and next to nearest-neighbors interactions, respectively, and they can be used to set the unit of energy. In our simulations, we fix $J_1 = J_2 = 1$ throughout —since $J_1, J_2 \geq 0$, we are dealing with the anti-ferromagnetic variant of the model . Note that $\mathcal{H}(J_1 \neq 0, J_2 = 0)$ is simply the famous XXZ Heisenberg chain. The terms $\lambda_1$ and $\lambda_2$ quantify the intensity of the interactions, which we choose $\lambda_1 = \lambda_2 = 0.55$ (this choice is also made, e.g., in [28]). Spin chains such as this one can be mapped exactly onto a spinless fermionic chain with an extra boundary term via the Jordan-Wigner transformation [53].

The crux of the matter of systems that reveal a MBL transition is that both their dynamical and static physical properties critically depend on the intensity of the disorder strength, $\omega$. Although the precise boundaries have not been completely delimited and in fact they vary depending on each particular model, the general scenario is more or less shared by all of them. While in the absence of disorder the XXZ chain is solvable by Bethe ansatz and thus integrable, the $J_1$-$J_2$ chain is considered as a paradigmatic model to exhibit quantum chaos and quantum thermalization [4]. For intermediate values of $\omega$, the chain shows an ergodic phase where most initial conditions are expected to thermalize. This region is generally said to be quantum chaotic in the sense that energy levels are characterized by level repulsion and the eigenvalue distribution is close to the Gaussian orthogonal ensemble (GOE) as in the predictions of RMT. This picture can be easily checked by statistical measures such as the nearest-neighbor spacing distribution (NNSD), $P(s)$, or the adjacent eigenlevel gap ratio, $P(r)$. However, as we

will show later, this metallic region hides long-range deviations from RMT universal results, which can be analyzed by computing the Thouless energy scale as was done in Refs. [23,54]. Dynamically, this region is dominated by sub-diffusive processes, multifractal scalings and other unexpected behavior [29–32]. The region in between the ergodic and the localized phases has been extensively studied. Recent results indicate that it may be characterized by a Griffiths-like phase in which anomalously different disorder regions seem to dominate the dynamics [33–35,55]; nonetheless, the debate is still open [56]. To describe the flow of intermediate statistics observed in this region, mean-field plasma models with effective power-law interactions between energy levels [24,57], the Rosenzweig-Porter ensemble with multifractal eigenvectors [58,59], a family of short-range plasma models [60] and generalizations [42,43] have been used. Finally, for disorder strengths larger than a critical value that is dependent on the dimension of the Hilbert space, the chain gradually reaches the MBL phase. Here, the ETH is violated, so generic initial conditions do not relax to their microcanonical average value. This region shows Poissonian spectral statistics instead, from which it follows that the spectrum behaves as a set of uncorrelated random numbers where level crossings can potentially take place [61]. This is explained by the identification of a complete set of local integrals of motion in the MBL phase [62,63]. Whether the MBL phase is reached in the thermodynamic limit from the ergodic phase in the form of an actual phase transition (i.e., by means of a critical point), or as a dynamical crossover, exhibiting a finite transition region (dubbed *bad metal*) with surviving non-ergodic but extended states is an open question [39,64–66]. The answer seems to be forever eluding the community due to the (strictly speaking) impossibility to access the thermodynamic limit and the importance of finite size effects [44,45], which are strong in these chains.

As the operator $\hat{S}^z := \sum_{\ell=1}^{L} \hat{S}_{\ell}^z$ commutes with the Hamiltonian, in this work we restrict our attention to the sector $S^z = 0$. The total dimension of the Hilbert space is then $d = \binom{L}{L/2}$ which grows asymptotically when $L \to \infty$ as $d \sim 2^L / \sqrt{\pi L/2}$. Thus, exact diagonalization with full calculation of all the eigenstates becomes realistic only up to chain lengths $16 \lesssim L \lesssim 18$ [67]. The spin chains that are frequently used to investigate the MBL phenomenon are known to exhibit strong border effects, meaning that the eigenstates are more localized as they get closer to the boundaries of the band, regardless of the disorder strength [37,39,68,69]. The question of whether true mobility edges survive in the thermodynamic limit in MBL systems is still open as they have also been argued to be indistinguishable from finite size effects [70]. In any case, to avoid border effects we will only consider the central $N = d/4$ eigenstates $\{|E_n\rangle\}_{n=1}^{N}$. The number of states we use range from $N = 63$ for $L = 10$ to $N = 3217$ for $L = 16$.

## 3 Extreme events across the transition

### 3.1 Quantum thermalization: the concept

Quantum thermalization refers to the equilibrium state that a quantum system reaches for sufficiently long times. Let us consider an isolated quantum system with Hamiltonian $H$ with eigenstates $\{|E_n\rangle\}_n$ satisfying $H|E_n\rangle = E_n|E_n\rangle$, and an arbitrary initial condition $|\psi(0)\rangle$. The latter evolves in time under $H$ and after an interval of $t \geq 0$ its wavefunction can be written $|\psi(t)\rangle = \exp(iHt/\hbar)|\psi(0)\rangle$. For a typical observable $\hat{O}$, one may consider the long-time average $\langle \hat{O} \rangle_t$ in the eigenbasis of $H$. If, for simplicity, the spectrum is non-degenerate, this is given by

$$\langle \hat{O} \rangle_t := \lim_{\tau \to \infty} \frac{1}{\tau} \int_0^{\tau} \mathrm{d}t \, \langle \psi(t)| \hat{O} |\psi(t)\rangle = \sum_n |C_n|^2 \langle E_n| \hat{O} |E_n\rangle \,, \tag{2}$$

where $C_n := \langle E_n | \psi(0) \rangle$ is a c-number linked to the probability of finding the system in the eigenstate $|E_n\rangle$. Under very relaxed assumptions [71, 72], long-time averages of the kind of Eq. (2) are known to reach a certain equilibrium value and remain close to it at all times. Nonetheless, the system is often said to *thermalize* only if that value corresponds to the particular case of the microcanonical average,

$$\langle \hat{O} \rangle_{\text{ME}} := \frac{1}{\mathcal{N}} \sum_{E_n \in [E-\Delta E, E+\Delta E]} \langle E_n | \hat{O} | E_n \rangle, \tag{3}$$

where $E$ is the macroscopic energy of the system and $\Delta E$ is a small energy window, $\Delta E / E \ll 1$, containing a large but finite number of levels, $1 \ll \mathcal{N} < \infty$. The connection between Eqs. (2) and (3) is well understood in classical dynamics; in particular, for classical chaotic systems, long-time averages are equivalent to phase averages if one fixes the right energy $E$, whereas integrable systems generically do not relax to Eq. (3) at all. In quantum mechanics, the eigenstate thermalization hypothesis [3–8] underlies the equivalence between these two averages. In particular, it states that the diagonal matrix elements of a typical observable $O_{nn} := \langle E_n | \hat{O} | E_n \rangle$ can always [8] be written

$$O_{nn} = \langle \hat{O} \rangle_{\text{ME}} + \Delta_n, \quad n \in \{1, \dots, N\}, \tag{4}$$

where $N$ denotes the Hilbert space size. The values $\Delta_n$ are called *diagonal fluctuations*, and describe the fluctuations of $O_{nn}$ around the microcanonical average $\langle \hat{O} \rangle_{\text{ME}}$. We can consider them as random numbers verifying $\langle \Delta_n \rangle = 0$ and $\langle \Delta_n^2 \rangle \neq 0$. The ETH states that a quantum system thermalizes if $\Delta_n$ (or, rather, its standard deviation, $\sigma_{\Delta_n} = \sqrt{\langle \Delta_n^2 \rangle}$) decreases fast enough with the Hilbert space dimension. In other words, the ETH is nothing more than a statement about how smoothly the diagonal terms must change with energy for a quantum system to reach the equilibrium value Eq. (3) for sufficiently long (but not exponentially long in the system size) times.

As indicated above, the standard tool to determine whether a quantum system thermalizes or not consists in studying if $\sigma_{\Delta_n}$ decreases fast enough with the system size. However, in addition to this fact, it has been recently shown that the presence of correlations in the $\Delta_n$ constitutes a signature of the existence of anomalous non-thermalizing initial conditions [54]. To better study these correlations with different observables, it is convenient to make the diagonal fluctuations in Eq. (4) dimensionless, by normalizing by its standard deviation, which yields

$$\widetilde{\Delta}_n := \frac{\Delta_n}{\sigma_{\Delta_n}} = \frac{O_{nn}}{\sigma_{\Delta_n}} - \frac{\langle \hat{O} \rangle_{\text{ME}}}{\sigma_{\Delta_n}}, \quad n \in \{1, \dots, N\}. \tag{5}$$

The consequence is that the new quantity in Eq. (5) is exactly a standard Gaussian random variable $\mathcal{G}(0, 1)$ with expected value 0 and variance 1, for any generic physical observable obeying RMT. Hence, as previous evidence strongly suggests that the breakdown of the ETH close to integrable regions may be a generic result [73–80], whereas thermalization is widely associated with ergodicity and and RMT, the quantity $\widetilde{\Delta}_n$ sharply discriminates between these two limiting regularity classes: for quantum chaotic systems it behaves as an uncorrelated, white random noise and is Gaussian, whereas for quantum integrable ones an emerging structure in its Fourier modes results from the existence of integrals of motion (i.e., the noise is no longer featureless). Thus, the (normalized) diagonal fluctuations $\widetilde{\Delta}_n$ provide a powerful tool to identify small deviations from thermalizing behavior, even in otherwise rather chaotic regions. In this direction, the power spectrum of $\widetilde{\Delta}_n$ has been recently used in conjunction with long-range spectral statistics to connect thermalization and Thouless energy in the prototypical disordered XXZ spin-1/2 chain [54].

## 3.2 Analysis of the diagonal fluctuations

Here, we study the diagonal fluctuations Eq. (5) across the MBL transition in the $J_1$-$J_2$ model, Eq. (1). As representative physical observables, we choose the full momentum distribution on a one-dimensional lattice with lattice constant set to unity, i.e.,

$$\hat{n}_q := \frac{1}{L} \sum_{m=1}^{L} \sum_{n=1}^{L} e^{2\pi i(m-n)q/L} \hat{S}_m^+ \hat{S}_n^-, \; q \in \{0, \ldots, L-1\}, \tag{6}$$

where $\hbar := 1$ and $\hat{S}_\ell^\pm$ are the usual ladder spin operators, which are related to those in Eq. (1) by the well-known expressions $\hat{S}_\ell^\pm = \hat{S}_\ell^x \pm i\hat{S}_\ell^y$. Thus, to calculate $\widetilde{\Delta}_n$, we first need to evaluate the diagonal matrix elements $O_{nn}$, for which the complete set of eigenstates $\{|E_n\rangle\}_n$ is needed. This is obtained by full diagonalization of Eq. (1). Next, the microcanonical average $\langle \hat{O} \rangle_{\mathrm{ME}}$ is obtained by fitting a polynomial of degree 4 to the previous matrix elements $O_{nn}$ to avoid the spurious effects originating in averages over finite energy windows [54, 81].

For reference, a summary of the number of central states $N = d/4$ and the number of realizations for each value of the number of sites $L$ and the disorder strength $\omega$ can be found in Table 1. These quantities have been chosen explicitly so that their product remains approximately constant. As not only the eigenlevels $\{E_n\}_n$ but also the eigenvectors $\{|E_n\rangle\}_n$ are needed, we cannot realistically increase the value of $L$ beyond 16. On the other hand, system sizes below $L = 10$ are indeed accessible but not very representative as the samples become statistically poor.

Table 1: Number of levels in the central region of the spectrum that is considered in the calculations $N = d/4$ and number of realizations for each value of the number of sites $L$ in the $J_1$-$J_2$ model, Eq. (1). The corresponding eigenvalues $\{E_n\}_{n=1}^N$ and eigenstates $\{|E_n\rangle\}_n$ have been obtained by exact diagonalization.

| $L$ | Levels in the central region | Realizations |
|---|---|---|
| 10 | 63 | 5000 |
| 12 | 231 | 1400 |
| 14 | 858 | 375 |
| 16 | 3217 | 100 |

We can now compute the probability distribution of such quantities $\widetilde{\Delta}_n$, $P(\widetilde{\Delta}_n)$. It is then straightforward to obtain the kurtosis excess which is directly related to the probability of *extreme events*. For a random variable $X$, the kurtosis excess $\gamma_2(X)$ is defined as

$$\gamma_2(X) := \mathrm{Kurt}[X] - 3 = \left\langle \left( \frac{X - \mu}{\sigma} \right)^4 \right\rangle - 3. \tag{7}$$

Here, $\mu := \langle X \rangle$ is the expected value of $X$ while $\sigma^2 := \langle X^2 \rangle - \langle X \rangle^2$ is its variance. The kurtosis of any univariate Gaussian distribution is 3, and thus we may use this quantity to compare how important is the presence of extreme events in a certain probability distribution with respect to a Gaussian. A positive kurtosis excess means that the distribution of $X$ is more tailed than a Gaussian while a negative kurtosis excess indicates the opposite. In what follows, the random variable of interest is $X = \widetilde{\Delta}_n$. The kurtosis excess of the quantity $X = O_{nn}$ was also calculated in Ref. [82] as a signature of anomalous thermalization.

In the main panel of Fig. 1 we show the kurtosis excess as a function of disorder for $L = 10, 12, 14, 16$. For small values of $\omega$, those corresponding to the ergodic phase of the model, the probability of extreme events roughly corresponds to the expected for a Gaussian distribution, $\gamma_2(\widetilde{\Delta}_n) \approx \gamma_2(\mathcal{G}) = 0$; the larger the chain size, $L$, the better the agreement. This means that the observables in Eq. (6) behave as expected in RMT in this region, and therefore ETH is fulfilled and thermalization is expected. Note that $\omega = 0$ has been excluded from our analysis as the clean system, a spin chain with periodic boundary conditions, is translationally invariant. The influence of these extra symmetries survives up to larger values of $\omega$ for smaller $L$ and $\gamma_2(\widetilde{\Delta}_n)$ increasingly deviates from 0 as a consequence. Then, the probability of extreme events as measured by $\gamma_2(\widetilde{\Delta}_n)$ shows a neat increase with disorder. However, this increase is not monotonic. On the contrary, $\gamma_2(\widetilde{\Delta}_n)$ has a maximum at a certain value of the disorder, which depends on $L$; this disorder strength will be denoted by $\omega_c(L)$ from now onward. In fact, we take the preceding statement as the *definition* of $\omega_c(L)$. After reaching this peak, $\gamma_2$ starts decreasing as a function of disorder, becoming negative for the largest values of disorder considered.

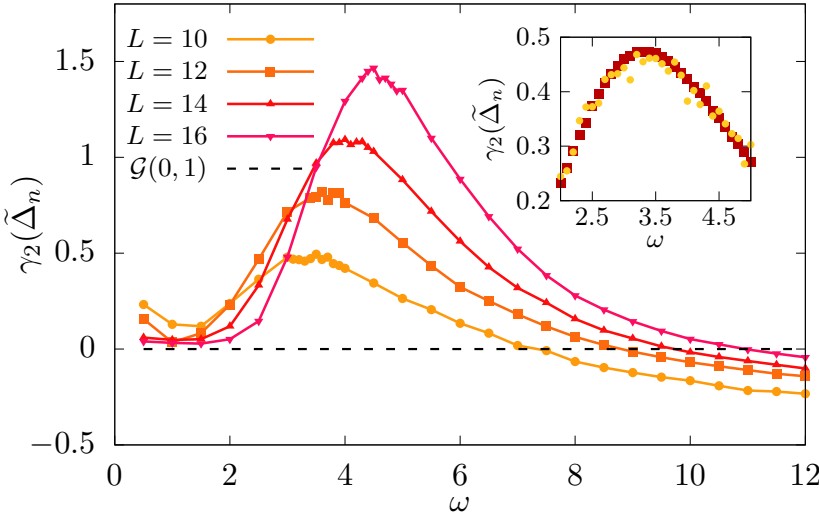

Figure 1: Main panel: kurtosis excess, $\gamma_2(\widetilde{\Delta}_n)$, Eq. (7), as a function of the disorder strength $\omega$ for the number of sites $L \in \{10, 12, 14, 16\}$. The black, dashed line represents the kurtosis excess for a standard Gaussian distribution $\mathcal{G}(0, 1)$, $\gamma_2(\mathcal{G}) = 0$. Inset: kurtosis excess for $L = 10$ obtained with $10^5$ realizations (red squares) and $10^3$ realizations (yellow circles). All results correspond to averages over all values of $q$.

These results suggest that the transition to the MBL phase starts with an increase of extreme events for the diagonal fluctuations, $\Delta_n$, which favor the existence of non-thermalizing initial conditions [54]. However, the integrable MBL phase is not characterized by a large probability for such extreme events, but by a large value of $\sigma_{\Delta_n}$ with a negative kurtosis excess. Hence, we can write a preliminary statement for the main conclusion of the paper: $\omega_c(L)$ *is a singular point in the transition from the ergodic to the MBL phase in the $J_1$-$J_2$ model, characterized by a maximum probability of extreme events for the diagonal fluctuations*.

From this result, one may ask about the actual shape of the distribution $P(\widetilde{\Delta}_n)$. This matter is addressed in Fig. 2, which consists of four panels. Here we fix the number of sites at $L = 16$. On the right-hand side we show $P(\widetilde{\Delta}_n)$ for three representative values of disorder. We also plot with black, dashed lines the probability density function of a Gaussian $\mathcal{G}(0, 1)$,

$P(x)_{\text{Gaussian}} := \exp(-x^2/2)/\sqrt{2\pi}$, $x \in \mathbb{R}$, which underlies this quantity in the case of thermalizing systems. For $\omega = 1$, we can see that this is indeed the case: $P(\widetilde{\Delta}_n)$ matches perfectly such a Gaussian distribution. It is worth to mention that no fitting has been performed here; we merely plot the Gaussian on top of the distributions. For $\omega = 4.5$, which is very close to the singular point $\omega_c(L = 16)$ (see next subsection), we observe that $P(\widetilde{\Delta}_n)$ no longer agrees with a Gaussian, and its tails display a slower decay, in concert with the high value of $\gamma_2(\widetilde{\Delta}_n)$ at this disorder. Instances of long tailed distributions and the breakdown of the ETH near the MBL transition are known, often associated with Griffiths effects [34, 35, 41, 54]. Finally, for a value of disorder well within the localized phase, $\omega = 100$, the distribution has been completely distorted and the tails decay faster than those of a Gaussian. To get a clearer picture, on the left-hand side of Fig. 2 we have chosen to plot the tails of the same distributions for $\widetilde{\Delta}_n \in [2, 6]$. This panel further confirms that the case $\omega = 4.5$ decays much more slowly than the others; as $\omega \to \infty$, the decay is faster and faster; and $\omega = 1$, matching very approximately the tails of the $\mathcal{G}(0, 1)$ distribution, seems to be in an intermediate situation.

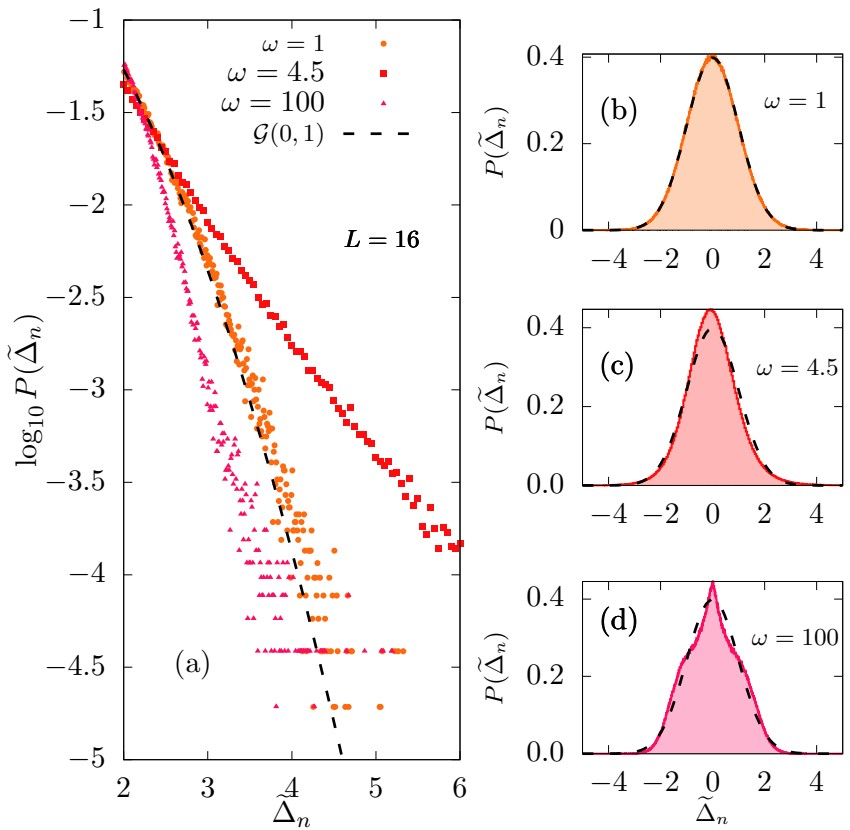

Figure 2: (a): Tails of the distribution of $\widetilde{\Delta}_n$, $P(\widetilde{\Delta}_n)$, for $L = 16$ and disorder values $\omega \in \{1, 4.5, 100\}$ in logarithmic scale. (b)-(d): probability distribution $P(\widetilde{\Delta}_n)$ for, from top to bottom, $\omega \in \{1, 4.5, 100\}$. The histograms bin has been chosen 0.05. Black, dashed lines represent the probability density function of a Gaussian $\mathcal{G}(0, 1)$.

## 3.3 Finite-size scaling

A precise calculation of $\omega_c(L)$ is a complicated task, requiring a huge number of realizations. To illustrate this fact, in the inset of Fig. 1 we plot the same results for $L = 10$, obtained with $10^3$ and $10^5$ realizations. The last curve is smooth and provides a pretty good value for

$\omega_c(L = 10)$. On the contrary, the curve obtained with $10^3$ realizations shows large fluctuations. As collecting so many realizations is too expensive for larger values of $L$, we rely on an heuristic ansatz for the position of the maximum,

$$\gamma_2(\omega) = \alpha |\omega - \beta|^{3/2} + \gamma. \tag{8}$$

We use this ansatz to fit the curves obtained with $10^5$ realizations for $L = 10$, shown in the inset of Fig. 1, and the ones for $L = 12$, $L = 14$ and $L = 16$ shown in the main panel of the same figure. From the results we infer how the position of the maximum, $\omega_c(L) \equiv \beta$, and its value, $\gamma_{2,\text{max}}(L) \equiv \gamma$, change with the system size. We collect all these results in the Fig. 3. We can see in panel (a) that the ansatz given in Eq. 8 provides a good description of the kurtosis excess, $\gamma_2$, around its maximum. We note, however, that this ansatz is not linked to critical exponents ($\alpha$ and $\gamma$ depend on the system size, so its shape is not universal); it is just an heuristic proposal to identify the position, $\omega_c(L)$, and the value, $\gamma_{2,\text{max}}(L)$, of the maximum of the kurtosis excess. Panel (b) shows the finite-size scaling of $\omega_c(L)$, which is well described by a linear law, $\omega_c(L) = \omega_0 + \omega_1 L$. We obtain $\omega_0 = 1.3$ and $\omega_1 = 0.2$. These results are compatible with the values obtained in Refs. [28, 47] from spectral statistics and entanglement entropy (they find $\omega_1 \approx 0.25$). It is worth to remark that this linear increase does not necessarily imply that $\omega_c(L) \to \infty$ in the thermodynamic limit. As discussed in [28, 47] the linear behavior may be only an approximation of a more complex behavior leading to a saturation of $\omega_c(L)$ when approaching the large-$L$ limit (see also Ref. [48] where an argument in the same direction was given in terms of phenomenological renormalization group flows [83]). Panel (c) shows that the same linear behavior is also found for the value of the maximum $\gamma_{2,\text{max}}(L) = \gamma_0 + \gamma_1 L$, with $\gamma_0 = -1.12$ and $\gamma_1 = 0.16$.

These results reinforce our previous conclusion. $\omega_c(L)$ may be associated with a precursor of a critical point in the ergodic-MBL transition. Unfortunately, much larger systems, far above the reach of current computers, are required to unveil the properties of this phase transition, or even whether it is an actual phase transition or just a smooth crossover. In any case, from the results discussed in this section, we can conjecture that, *if there exists a critical point, $\omega_c(\infty)$, then it must be the one in which $\gamma_2(\widetilde{\Delta}_n)$ is maximum*. In the next section we will show that the singular character of $\omega_c(L)$ is fully compatible with the behavior of spectral statistics across the transition, widely used to characterize the static properties of the chain.

## 4 Spectral statistics across the transition

The statistical analysis of eigenlevel fluctuations is by now well established as one of the main tools to study quantum complex systems. This is because the spectral properties of quantum systems with a chaotic classical counterpart do not depend on the particular features of the Hamiltonian but only on its more general global symmetries. As a result, the level statistics of a variety of apparently completely unrelated physical systems universally coincide with the predictions of RMT for the random matrix ensemble associated to their particular symmetry [10].

The eigenlevel distribution of the kind of spin chains associated to the MBL transition has been abundantly studied by many previous works. The goal of this section is to show that the maximum of the probability of extreme events as measured by the kurtosis in Fig. 1 acts as an indicator of the value of disorder beyond which the chaotic regime is completely abandoned and the transition to the MBL phase begins. We will show that the spectral statistics are not only quantitatively but also qualitatively different on both sides of this point. For $\omega \lesssim \omega_c(L)$, the system belongs in the chaotic region where the Thouless energy, the scale associated with RMT correlations, is larger than the Heisenberg energy, the mean energy distance between

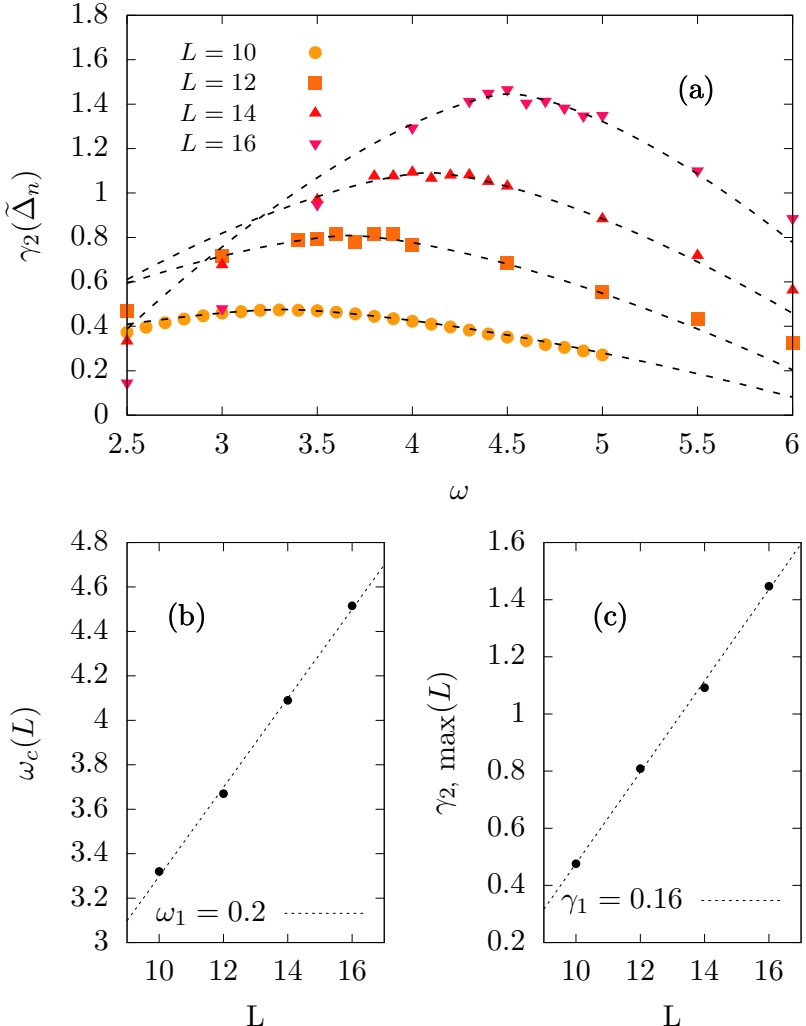

Figure 3: (a) Kurtosis excess, $\gamma_2(\widetilde{\Delta}_n)$, Eq. (7), as a function of the disorder strength $\omega$ for the number of sites $L \in \{10, 12, 14, 16\}$. The dashed line shows the fit to Eq. 8. (b) Critical value, $\omega_c(L)$, as a function of the system size, $L$. The dotted line displays the best linear fit, $\omega_c(L) = \omega_0 + \omega_1 L$. (c) Maximum of the kurtosis excess, $\gamma_{2,\text{max}}(L)$, as a function of $L$. Again, the dotted line displays the best linear fit, $\gamma_{2,\text{max}}(L) = \gamma_0 + \gamma_1 L$.

levels. We focus here in the case with $L = 16$, for which long-range spectral statistics give a better result. We leave for Sec. 5 the finite-size scaling.

Here we make use of two different measures: the NNSD, $P(s)$, which is concerned with short-range level correlations, and the $\delta_n$, which is a long-range spectral statistic. For the universal predictions of RMT to hold it is necessary to obtain the cumulative spectral function, $G(E) = \sum_{n=1}^{N} \Theta(E - E_n)$, where $\Theta$ is the unit step function, representing the number of levels with energy less than or equal to a certain value $E$ [9–11]. This function can be separated into a smooth part $\overline{G}$ and a fluctuating part, $\widetilde{G}$, as $G(E) = \overline{G}(E) + \widetilde{G}(E)$. Then, the original eigenlevels $\{E_n\}_{n=1}^{N}$ are mapped onto the dimensionless quantities $\{\varepsilon_n\}_{n=1}^{N}$ as $E_n \mapsto \overline{G}(E_n) =: \varepsilon_n$. As a consequence, the mean level density is unity, and RMT results can be applied. This procedure is termed *unfolding* [81, 84]. Since there is no statistical theory for Eq. 1 that provides a theoretical expression for the smooth $\overline{G}(E)$, in this work we numerically obtain it by fitting a

polynomial of degree 10 to the original energies $\{E_n\}_{n=1}^N$, i.e., $\overline{G}(x) = \sum_{k=0}^{10} a_k x^k$.

The famous NNSD, $P(s)$, is the distribution of the (unfolded) consecutive level spacings, $s_i := \varepsilon_{i+1} - \varepsilon_i \geq 0$, that is, $P(s) := \langle \delta(s - s_i) \rangle$. For systems that are invariant under orthogonal transformations, the eigenlevel distribution follows the results for GOE random matrices, and quantum chaos manifests via level repulsion in the Wigner-Dyson surmise $P(s) = \frac{\pi}{2} s \exp\left(-\pi s^2/4\right)$. By contrast, the spectrum of generic quantum integrable systems is equivalent to a set of independent, identically distributed, Poisson random variables [61] where level repulsion is no longer present, and thus $P(s) = \exp(-s)$.

Long-range statistics describe the spectral properties of eigenlevels separated by large energy index distances (i.e., levels $E_i, E_j \in \{E_n\}_{n=1}^N$ with $|i-j| \lesssim N$), as opposed to level distances of one or two units (or in general $|i-j| \ll N$). Contrary to short-range statistics, these allow to obtain the so-called Thouless energy scale, $E_{\text{Th}}$: the energy scale beyond which universal RMT results break down [85, 86]. It has been recently shown that, at least in the disordered XXZ spin chain, this scale determines to what extent a given system thermalizes [54]. A decreasing value of $E_{\text{Th}}$ was linked to an increase of the probability of extreme events from the ergodic region. RMT results have been shown to describe well the statistics of eigenvalues separated by less than $E_{\text{Th}}$; however, for level distances larger than $E_{\text{Th}}$, long-range deviations towards the corresponding integrable result can be detected even in the region interpreted as chaotic by short-range statistics such as the $P(s)$. In the case of disordered spin chains, the number variance [23], the spectral form factor [45], and very recently the $\delta_n$ [54] have allowed to calculate the value of $E_{\text{Th}}$.

In this work we will use the $\delta_n$ spectral statistic as a long-range measure of level correlations [87–92]. It is defined as the difference between the $n$th unfolded level and the corresponding energy value in an equiespaced spectrum (i.e., with $\langle \varepsilon_n \rangle = n$), that is,

$$\delta_n := \varepsilon_n - n, \quad n \in \{1, \ldots, N\}. \tag{9}$$

The quantity $\delta_n$ can be understood as a discrete time series where the level order index $n$ plays the role of a discrete time. Thus a discrete Fourier transform can be applied to $\delta_n$, one of the most common techniques in time series analysis. Taking its square modulus gives the power spectrum $P_k^{\delta}$ of the signal. The quantity of interest is the averaged power spectrum, which reads

$$\langle P_k^{\delta} \rangle := \langle |\mathcal{F}(\delta_n)|^2 \rangle = \left\langle \left| \frac{1}{\sqrt{N}} \sum_{n=1}^N \delta_n \exp\left(\frac{-2\pi i k n}{N}\right) \right|^2 \right\rangle, \quad k \in \{1, 2, \ldots, k_{\text{Ny}}\}, \tag{10}$$

where $k_{\text{Ny}} := N/2$ is the Nyquist frequency. Here the angular brackets $\langle \cdot \rangle$ denote average over realizations for a fixed disorder strength. In the limit $k/N \ll 1$ and $N \gg 1$, the power spectrum of $\delta_n$ in quantum integrable systems exhibits the neat power-law decay $\langle P_k^{\delta} \rangle \simeq 1/k^2$, whereas for quantum chaotic ones this is $\langle P_k^{\delta} \rangle \simeq 1/k$ [87]. One says that this is a universal feature because it is only dependent on the regularity class (integrable or chaotic) of the Hamiltonian matrix; however, it does not depend on the underlying symmetry.

## 4.1 Semi-Poisson model for short- and long-range spectral statistics

While a quantum chaotic spectrum exhibits correlations between levels, these are completely absent from quantum integrable ones (i.e., levels are uncorrelated). Wigner-Dyson and Poisson statistics are valid, respectively, in the limiting metallic and insulator regimes of the metal-insulator transition (MIT) of the Anderson model [93]. At the MIT critical point a new universal model of spectral statistics is believed to be applicable, the semi-Poisson [86], which is connected to the fractal nature of the critical wavefunctions, intermediate between the extended and localized cases [57, 94]. Semi-Poisson spectra are intermediate between the two

opposed regimes of Wigner-Dyson and Poisson in the sense that they reveal some degree of level repulsion but for asymptotically large level distances the NNSD decreases much more slowly than for GOE spectra, producing $P(s) = 4s \exp(-2s)$ [95,96].

The semi-Poisson statistics can be obtained from a short-range plasma model in which the particles play the role of the energy levels, interacting only with their corresponding nearest neighbours [95,97–100]. The result is a family of distribution of independent level spacings,

$$P(s;\eta) := \frac{\eta^\eta s^{\eta-1} e^{-\eta s}}{\Gamma(\eta)}, \ s \geq 0, \ \eta \in [1,+\infty), \tag{11}$$

where $\Gamma(z) = \int_0^\infty \mathrm{d}t \, t^{z-1} e^{-t}$ is the gamma function. The case with $\eta = 1$ corresponds to a Poissonian spectrum. If $\eta > 1$, the NNSD reveals level repulsion proportional to $P(s) \propto s^{\eta-1}$, but the fall-off $\exp(-\eta s)$ is always slower than $\exp(-s^2)$ in the Wigner-Dyson surmise of chaotic systems.

The power spectrum of $\delta_n$ is known for the family of distributions in Eq. (11). The *exact* result for a set of $N$ uncorrelated spacings after normalizing by the estimator of the level spacing mean [84] is given by

$$\langle P_k^\delta \rangle(\eta) := \left( \frac{N}{\eta N + 1} \right) \frac{1}{4 \sin^2(\omega_k/2)}, \ \omega_k := \frac{2\pi k}{N+1}, \ k \in \{1, 2, \ldots, N+1\}. \tag{12}$$

Equation (12) depends on the single continuous parameter $\eta \in [1,+\infty)$, with $\eta = 1$ again corresponding to Poissonian statistics, and $\eta = 2$ being semi-Poisson in the strictest sense of the term. From Eq. (12) it follows immediately that the crossover between these two limits takes the form of a vertical translation of $\langle P_k^\delta \rangle$ but its overall structure remains unchanged. In the MIT transition this generalized semi-Poisson model seems to describe the spectral statistics of the critical region with a value of $\eta$ that changes from 2 to 1 as the dimensionality is increased as has been numerically investigated up to six spatial dimensions [101].

Therefore, Eq. (11) and Eq. (12) allow to completely characterize a crossover from semi-Poissonian to Poissonian statistics. Although the semi-Poissonian limit strictly refers to the case $\eta = 2$ alone, one usually still uses the term to allude to the whole family of distributions. As the value of $\eta$ is reduced the family of distributions exhibit a decreasing intensity of level repulsion, which completely vanishes only at the Poissonian limit.

Here we show that the semi-Poisson model provides a characterization of the level statistics for $\omega \gtrsim \omega_c(L)$ in Eq. (1), for both short- and long-range correlations. To that end, first we have computed the NNSD for the eigenlevels of our model, Eq. (1). We have unfolded the sequence of original energies $\{E_n\}_{n=1}^N$ with a polynomial of degree 10. Starting from the central $d/3$ levels, we have removed the $2\lfloor d/48 \rfloor$ levels closest to the edges before and after unfolding, which gives $N = \lfloor d/4 \rfloor$ for our analysis (see Table 1). To avoid the spurious effects introduced by spectral unfolding in systems close to integrable dynamics, we have divided each level spacing by its mean value for each realization, so that the mean level spacing becomes $\langle s \rangle = 1$ exactly [84]. Then, we have constructed the histograms of $P(s)$ with a bin size $\mathrm{d}s = 0.1$. After that, we performed a single-parameter fit of Eq. (11) to the $P(s)$ to obtain the value of $\eta$. Results are shown in Fig. 4 for a set of disorder values and $L = 16$, for which Eq. (11) provides a good description. Below $\omega = 4.7$, the case displayed in panel (a) of this figure, results are not so good. Hence, we take this value as an estimate of the critical point. It is very close to the one inferred from Fig. 3, $\omega_c(16) = 4.52$. We leave a direct comparison between both estimates to Sec. 5.

As can be seen, all the numerical histograms shown in Fig. 4 $P(s)$ are very well described by Eq. (11) for disorder values *larger* than $\omega_c(16)$. Incidentally, for $\omega = 5.0$, we obtain almost fully semi-Poissonian level spacings, $\eta \approx 2.079$. For $\omega = 4.7$, we obtain $\eta \approx 2.274$, implying that the level repulsion is stronger that in the ergodic case. However, it is difficult to infer

whether this is a spurious result due to the proximity to the singular point, $\omega_c(L)$, or if it is just a finite-size effect. As $\omega$ is increased into the localized phase, the parameter $\eta$ decreases, until it reaches a value of $\eta \approx 1.058$ for $\omega = 12$. Well within the localized phase, the NNSD agrees well with $\eta \approx 1$ (not shown). For $\omega$ smaller than $\omega = 4.7$, the system enters the chaotic region. Equation (11) is explicitly derived on the assumption of statistically independent spacings as in the Berry-Tabor result [61]. Then, since quantum chaotic eigenlevels are statistically correlated, Eq. (11) cannot satisfactorily describe this side of the transition. However, for $\omega \gtrsim 4.7$, *level spacings approximately become independent random variables*, but as we have seen they differ from (generic) integrable systems in that their distribution is not Poissonian.

This picture is also obtained in long-range measures of level statistics. This means that this side of the transition can be correctly described by spectral statistics ranging from semi-Poisson to Poisson for eigenenergies separated by not only short but also large level distances.

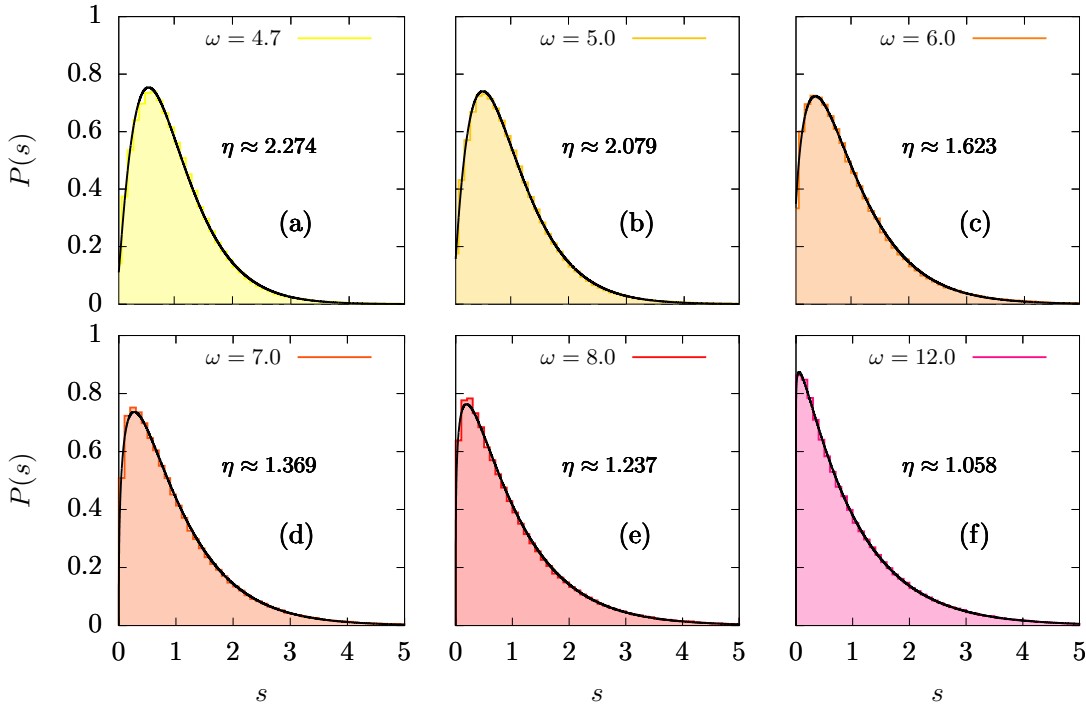

Figure 4: (a)-(f) Nearest-neighbor spacing distribution, $P(s)$, for the six values of disorder strength $\omega \in \{4.7, 5.0, 6.0, 7.0, 8.0, 12.0\}$ (colour filled histograms). Solid, black lines represent the best nonlinear fit of Eq. (11) to the histograms of $P(s)$. Bin size has been chosen $ds = 0.1$. Results correspond to $L = 16$.

In Fig. 5 we show the numerically obtained power spectrum $\langle P_k^\delta \rangle$ together with Eq. (12) for the values of $\eta$ from the NNSD for the same values of the disorder strength. The Poisson and GOE results are also shown for reference. For brevity we omit the analytic expression of the GOE but it can be found in Ref. [88]. For $\omega \gtrsim 5.0$, the agreement between the two curves is almost perfect. Below this disorder strength, for $\omega = 4.7$, the power spectrum is slightly touching the GOE result. This reflects that this value of $\omega$ can indeed be taken as the limit beyond which the model of statistically independent spacings is valid. As $\omega$ is increased, $\eta$ decreases towards the Poisson result, and the power spectrum undergoes a smooth crossover, approaching the theoretical Poisson curve vertically. We stress that no fitting has been performed in this case; the black solid lines merely correspond to Eq. (12) for $\eta$ as obtained in Fig. 4, for each value of $\omega$ there shown. The first frequencies are spoiled in all six cases, but this is an expected consequence from the unfolding procedure [81], not a shortcoming of our

model.

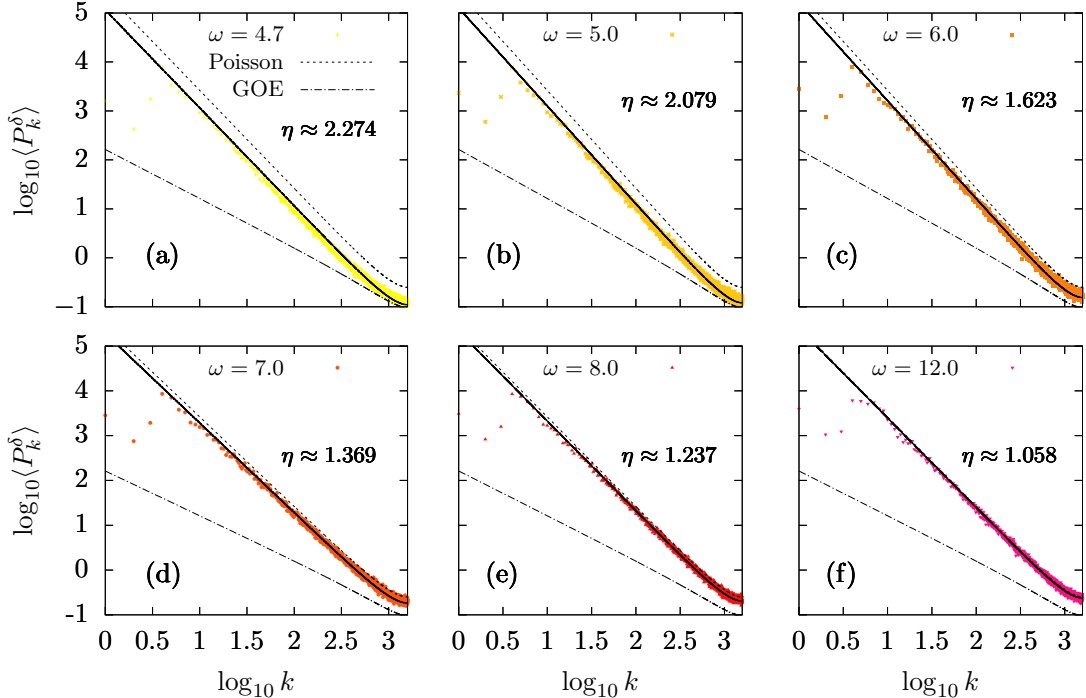

Figure 5: (a)-(f) Power spectrum of $\delta_n$, $\langle P_k^\delta \rangle$, for the six values of disorder strength $\omega \in \{4.7, 5.0, 6.0, 7.0, 8.0, 12.0\}$ (color points). Solid, black lines represent Eq. (12) for the values of $\eta$ obtained from the NNSD, Fig. 4. Top and bottom dashed lines are the Poisson [Eq. (12) with $\eta = 1$] and GOE results (see Ref. [88]) respectively. Results correspond to $L = 16$.

## 4.2 Long-range spectral statistics on the ergodic region

Short-range spectral statistics in the chaotic region show results consistent with GOE random matrices as has been extensively shown before (see, e.g., [39]), so we will not consider them further. Additional investigation of the long-range spectral correlations across the transition is provided in Fig. 6, where we focus on the chaotic side of the model Eq. (1) instead. There we show the numerical $\langle P_k^\delta \rangle$, obtained following the same steps as before. For $\omega = 0.5$ and $\omega = 1.0$, the differences between the chaotic theoretical curve and the numerics are minimal. However, as $\omega$ is increased, the power spectrum gradually deviates from the ergodic towards the integrable curve. This effect is blurred in panels (a) and (b), due to the unfolding procedure [81], but it is clearly seen in the rest of the panels. This deviation is linked to the Thouless energy $E_{\text{Th}}$, the energy scale beyond which energy levels are no longer correlated like in RMT. Following the same procedure that in [54], the power spectrum of $\delta_n$ gives us direct access to the Thouless frequency $k_{\text{Th}}$, which determines a characteristic length $\ell_{\text{Th}} = N/k_{\text{Th}}$: two energy levels, $E_n$ and $E_m$ are correlated like in RMT if their level index distance satisfies $|n-m| < \ell_{\text{Th}}$. In this sense, a fully chaotic RMT spectrum is one that has the highest possible value of $\ell_{\text{Th}}$ (or the lowest possible value of $k_{\text{Th}}$). This would indicate that GOE correlations are shared by levels separated by any distance within the spectrum boundaries. As $k_{\text{Th}}$ increases, the spectrum is thus 'less chaotic' in this particular sense[1]. A good estimation for $k_{\text{Th}}$ is to choose

---

[1]One may in fact estimate the Thouless energy as $E_{\text{Th}} = \hbar/\tau_{\text{Th}} = \ell_{\text{Th}}/g(\epsilon)$, where where $g(\epsilon)$ is the density of states at the average energy. Thus, $E_{\text{Th}} \propto \ell_{\text{Th}} \approx \ell_{\text{max}} \propto k_{\text{min}}^{-1}$. Note that $E_{\text{Th}}$ and $\tau_{\text{Th}}$ have dimensions of energy

the lowest possible frequency for which $\langle P_k^\delta \rangle$ fluctuates *below* the GOE curve, which gives the approximation $k_{\min} \approx k_{\text{Th}}$. This point is identified in all the panels of Fig. 6 by means of a vertical arrow. We can see that $k_{\min}$ monotonically increases with $\omega$, showing that the system becomes less chaotic as the singular point, $\omega_c(L)$, is approached. It is worth to note that this degree of detail cannot be achieved by analyzing short-range spectral statistics, even in the case where $k_{\min}$ is very large. Because they are by definition insensitive to the spectral properties of distant levels, short-range statistics would still produce a rather chaotic result (this can be seen, e.g., in the mean value of the adjacent level gap ratio, sometimes employed for finite-size scaling considerations).

### 4.3 Transition landscape

Results plotted in Figs. 5 and 6 suggest a scenario summarized in Fig. 7. In panel (a) of this figure, we display $k_{\min}$ as a function of $\omega$. We observe that the value of $k_{\min}$ for $\omega \ll \omega_c(L)$ is very small compared to the number of levels, $k_{\min}/N \ll 1$, which gives a large value of the Thouless energy. It is seen that $k_{\min}$ then grows quite fast with $\omega$, explaining the subsequent separation from the GOE curve of the power spectrum for those values of disorder. The limiting value $k_{\min} = k_{\text{Ny}} = N/2$ is also shown with a dashed line in panel (a) of Fig. 7. Reaching the Nyquist frequency indicates that the power spectrum has completely separated from the GOE curve, and hence the quantum correlations of RMT are destroyed at all scales (i.e., there is no level index distance such that levels separated by that distance are correlated). We can see an abrupt jump towards the Nyquist frequency at $\omega \approx 4.7$, supporting our previous claim that this value constitutes a good estimate of the critical disorder strength. *This means that* $\langle P_k^\delta \rangle$ *completely separates from the RMT result at a value of the disorder compatible with* $\omega_c(L)$.

From this point onward, the power spectrum is characterized by Eq. (12) instead because level correlations are no longer present. This is shown in panel (b) of Fig. 7, where we display $\eta$ as a function of $\omega$. In concert with previous Fig. 1, $\eta$ smoothly decreases as the MBL phase is approached. Furthermore, a value close to $\eta = 2$, which implies a level repulsion equal to the GOE, is found around the singular point $\omega_c(L)$. As we have discussed before, our results are compatible with a *strange* singular point $\omega_c(L)$ characterized by semi-Poisson statistics with level repulsion larger than in the GOE. Notwithstanding, this may be also a finite-size effect. In any case, we find a neat transition in the spectral statistics between full or partial correlations with level repulsion [the chaotic region, $0.5 \lesssim \omega \lesssim \omega_c(L)$], to no correlations but still level repulsion [semi-Poisson region, $\omega_c(L) \lesssim \omega < \infty$], to finally no correlations and no level repulsion whatsoever (Poisson limit, $\omega \to \infty$).

All these results reinforce our previous conclusion regarding the singular character of $\omega_c(L)$. Furthermore, *the disorder value at which the kurtosis of the diagonal fluctuations,* $\gamma_2(\widetilde{\Delta}_n)$*, is maximum ,* $\omega_c(L)$*, is also a singular point regarding spectral statistics.*

## 5 Discussion of results

The results from previous Secs. 3 and 4 provide an interesting insight into the mechanism for the transition between the chaotic and the localized phases in the $J_1$-$J_2$ model. It can be summarized as a *conjecture* as follows:

*If there exists a critical point in the transition from the ergodic to the MBL phases,* $\omega_c(L \to \infty)$*, then it must correspond to the disorder strength at which the kurtosis of the distribution of the (normalized) diagonal fluctuations* $\gamma_2(\widetilde{\Delta}_n)$ *is maximum . For finite systems,* $\omega_c(L < \infty)$ *splits*

and time, respectively, but $\ell_{\text{Th}}$ is a dimensionless quantity that refers to the unfolded level distance; the 'frequency' $k_{\text{Th}}$ is also dimensionless.

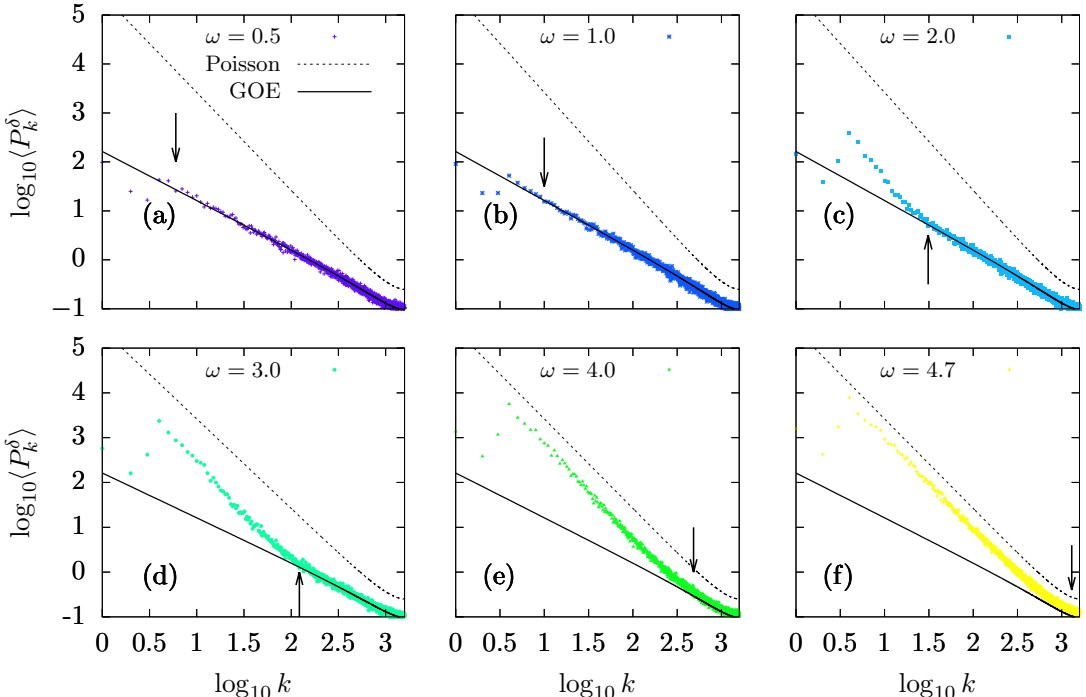

Figure 6: (a)-(f) Power spectrum of $\delta_n$, $\langle P_k^\delta \rangle$, for six values of disorder strength $\omega \in \{0.5, 1.0, 2.0, 3.0, 4.0, 4.7\}$ (color points). Top and bottom black lines are the Poisson [Eq. (12) with $\eta = 1$] and GOE results (see Ref. [88]), respectively. The arrows in each panel indicate the value of $k_{\min}$ for the corresponding value of $\omega$. All results correspond to $L = 16$.

*the chaotic and localized phases.*

As this conjecture seems based on just the approximate coincidence between the critical disorder strength for $L = 16$, estimated from both the kurtosis excess, $\omega_c(16) = 4.52$, and spectral statistics, $\omega = 4.7$, we show in Fig. 8 an integrated scenario. In Fig. 8(a), we show the re-scaled kurtosis excess, $\gamma_2(L)/\gamma_{2,\max} = \gamma_2(L)/(\gamma_0 + \gamma_1 L)$, as a function of $\omega - \omega_c(L) = \omega - \omega_0 - \omega_1 L$, for $L = 10, 12, 14$, and 16. We can see that the results for these four different system sizes collapse onto a single curve around the singular point $\omega = \omega_c(L)$. This suggest that the transition from the ergodic to the MBL phases shows the same features from $L = 10$ to $L = 16$, although the maximum of the kurtosis excess increases with the system size. In Fig. 8(b), we perform a similar finite-size scaling analysis for the spectral statistics. In particular, we display the distance between the numerics obtained for the NNSD and the family of generalized intermediate statistics, given by Eq. (11),

$$\Delta_{\text{SP}} = \frac{1}{N_b} \sum_{i=1}^{N_b} |P_H(s_i) - P(s_i; \eta)|^2 \,, \tag{13}$$

where $P_H(s)$ represents the numerical results, $P(s; \eta)$ is Eq. (11), and $N_b$ is the total number of bins used to build the histogram. All the calculations are done with a bin size $ds = 0.1$, and the fit is performed over $0 < s_i < 5$. Exactly as in Fig. 8(a), the distance $\Delta_{\text{SP}}(L)$ is plotted versus $\omega - \omega_c(L) = \omega - \omega_0 - \omega_1 L$, that is, *relying on the estimate of the critical point inferred from the kurtosis excess, displayed in Fig. 3*. We can see that the results for $L = 12, 14$ and 16 collapse very approximately onto a single curve, and that $\omega = \omega_c(L)$ constitutes a very good estimate of the disorder strength above which the generalized semi-Poisson distribution,

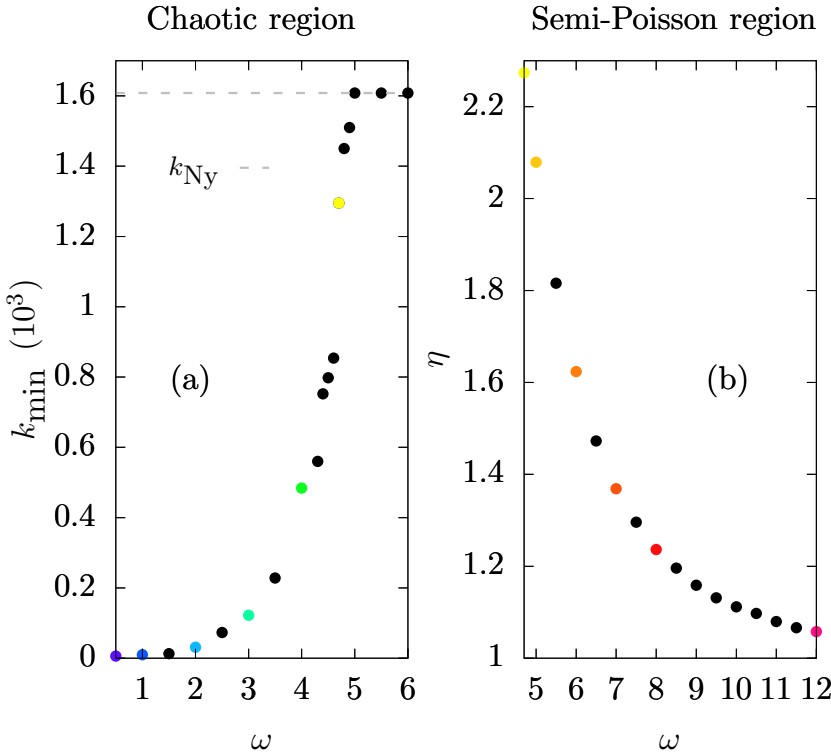

Figure 7: Behavior of long-range spectral statistics as a function of disorder $\omega$ for $L = 16$ across the transition. (a): For small values of disorder, $\omega \lesssim \omega_c(L = 16)$, we represent the characteristic frequency $k_{\min}$ as a function of $\omega$ and compare to the Nyquist frequency $k_{\text{Ny}} = N/2$ (gray, dashed line). A small $k_{\min}$ corresponds to a large Thouless energy and vice versa. (b): For large values of disorder, $\omega \gtrsim \omega_c(L = 16)$, we plot $\eta$ obtained from a single-parameter fit of the generalized semi-Poisson distribution Eq. (11) to the numerically obtained NNSD ($\eta = 1$ corresponds to the full Poisson limit). For convenience of the reader, colored points represent the values of $k_{\min}$ and $\eta$ for the same values of disorder and color code as in Figs. 4, 5, and 6.

Eq. (11), provides an accurate description of the numerical results[2]. These results provide a strong support for our previous conjecture. The singular point, $\omega_c(L)$, splits the behavior of the system into two different dynamical phases:

- *Chaotic phase.* This corresponds to small values of disorder, $\omega < \omega_c(L)$. Here spectral statistics coincide with RMT up to a certain characteristic length $\ell_{\max}$, beyond which RMT-like correlations are lost [54]. For very small values of $\omega$, the ETH is fulfilled, and generic observables relax to their microcanonical equilibrium value. As can be seen in Fig. 1, the probability of extreme events is very approximately the same than for a Gaussian distribution, which underlies the diagonal matrix elements for thermalizing systems. Even more, panels (a) and (b) of Fig. 2 show that $\widetilde{\Delta}_n$ agrees almost perfectly with such a distribution, including in the tails. As $\omega$ is increased, the kurtosis excess $\gamma_2(\widetilde{\Delta}_n)$ increases too, significantly separating from the Gaussian expectation. As was shown in Ref. [54], this means that, although generic initial conditions will thermalize to the microcanonical average for these disorder strengths, it is also a lot more probable

---

[2]The case with $L = 10$ is not considered, because it is well known that the behavior of short-range spectral statistics is qualitatively different for $L < 12$ along the whole MBL transition [15]

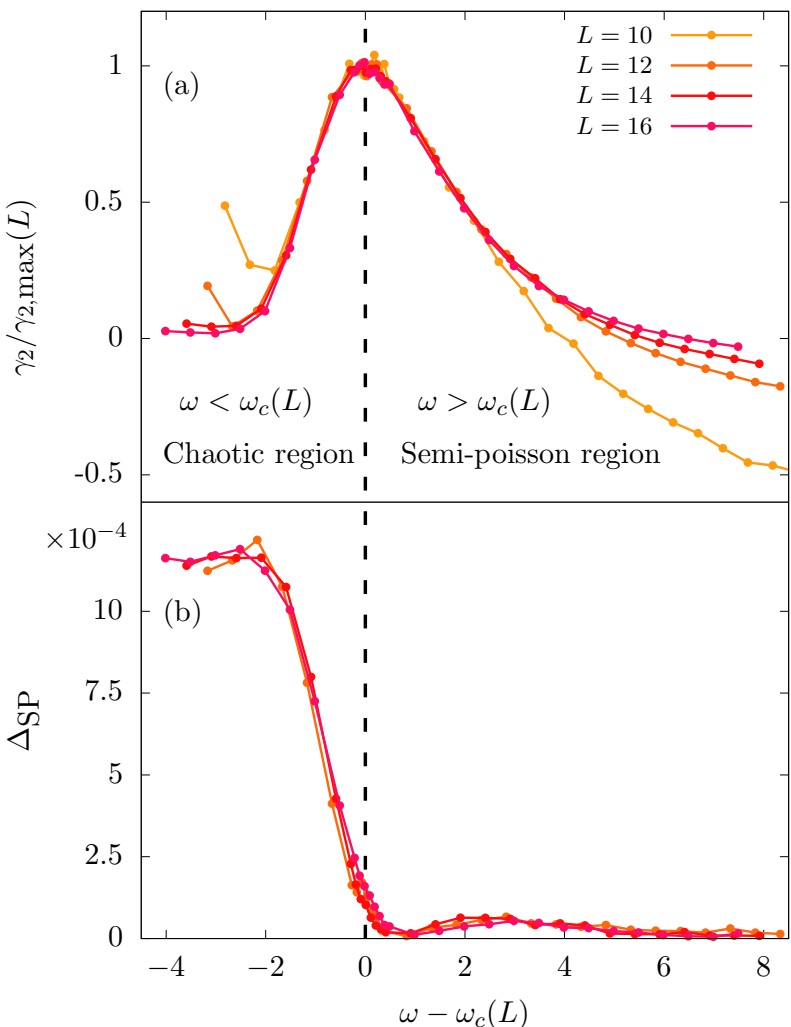

Figure 8: (a): Kurtosis excess of the distribution of $\widetilde{\Delta}_n$, re-scaled with its maximum value for each system size, $\gamma_{2,\text{max}}$, as a function of $\omega - \omega_c(L)$, for $L = 10, 12, 14$, and 16. The black, dashed line shows the singular point, $\omega = \omega_c(L)$. (b): Distance between the numerical NNSD and the family of intermediate statistics Eq. (11), $\Delta_{\text{SP}}$, as a function of $\omega - \omega_c(L)$, for $L = 12, 14$, and 16.

to find anomalous initial conditions that do not. In terms of spectral statistics, even though for small $\omega$ the eigenlevel distribution is very close to the GOE predictions of RMT, Fig. 6 shows that as $\omega$ increases (but still remains on the chaotic side) long-range deviations can be attributed to the anomalous behavior of the Thouless energy. The increasing values of the probability of extreme events are thus connected to the gradual loss of chaos in the spectrum. For small values of $\omega$, $\widetilde{\Delta}_n$ is Gaussian, the system thermalizes, and GOE level correlations are maintained between levels separated by distances comparable to the total size of the spectrum. As $\omega$ increases, extreme events take place with more and more probability and the Thouless energy starts decreasing, meaning that level correlations between levels very far apart from each other are being destroyed. For disorder values close to $\omega \approx \omega_c(L)$, the Thouless energy is minimal and the model departs from its chaotic phase.

- *Semi-Poisson phase.* This corresponds to $\omega > \omega_c(L)$. At this point the probability of extreme events of Fig. 1 starts diminishing (for $L = 16$); at around $\omega \approx 11$, it crosses the corresponding value for a Gaussian distribution. As can be seen in panels (a) and (d) of Fig. 2, well within the localized phase, at $\omega = 100$, the distribution of the diagonal fluctuations has been completely distorted and for asymptotically large values it decreases faster than a Gaussian distribution, in concert with the finding that the probability of extreme events is smaller than that of a Gaussian for large disorder. On this side of the transition, Eqs. (11) and (12) account for both short and long-range spectral statistics, as can be seen in Figs. 4 and 5. This means that the spectrum is here approximately composed of independent, identically distributed random numbers that still show level repulsion, so they are intermediate between GOE and Poisson. For $L < \infty$, the Poisson limit is only strictly reached when $\omega \to \infty$.

The singular point separating these two dynamical phases, $\omega_c(L)$, shows the following features. First, it is the disorder strength for which the maximum probability of extreme events in the diagonal fluctuations occurs. Here $\widetilde{\Delta}_n$ is no longer well described by a Gaussian, and the decay of its tails is much slower, almost exponential as panel (a) of Fig. 2 suggests. Second, it indicates certain singularity in the spectral statistics in the sense that below it level correlations exists between levels separated by certain distances, but beyond it no such feature can be found, even though some degree of level repulsion is still preserved. This scenario is compatible with a two-stage transition, in concert with previous numerical findings [24, 30]. And third, it linearly increases with the system size, $\omega_c(L) = \omega_0 + \omega_1 L$, at least for system sizes small enough to be exactly diagonalized, and both for the kurtosis excess and the NNSD distribution. This scaling law is compatible with the recent results published in [28, 47], where the transition is identified to be in the Berezinskii-Kosterlitz-Thouless class [102, 103].

In Ref. [30] the transition from the ergodic to the MBL phase was identified by means of a nonuniversal jump of the multifractal dimensions (both in Fock and spin configuration basis). We find that a similar effect gives rise to a maximum value of $\gamma_2(\widetilde{\Delta}_n)$, which also hints towards the existence of a critical transition point in the $J_1$-$J_2$ model as we have shown. The multifractal dimensions vanish only in the infinite disorder limit, coinciding with full Poissonian statistics where level repulsion completely vanishes as well since $\eta = 1$ [see Eq. (11)]. Our results are also consistent with those presented in Ref. [24], where the effective interaction between eigenlevels in the disordered XXZ spin chain was analyzed. In the ergodic phase, level statistics were characterized by a long-range plasma model. However, upon reaching the MBL transition, a power-law local interaction between levels means that these are intermediate between Wigner-Dyson and Poisson, leading to the family of semi-Poisson distributions as we have seen. The locality of interactions on this side of the transition is also consistent with the level repulsion $P(s) \propto s^{\eta-1}$, which becomes increasingly weaker as we approach the Poisson limit and hence the interaction between level spacings also diminishes up to this point.

Finally, we wish to emphasize that other more common signatures of the transition from ergodicity to MBL, like the mean value of the adjacent level gap ratio or the family of Rényi entropies (of which the Shannon entropy is a particular case) [15, 23, 24, 36–43], change monotonically with $\omega$, and therefore do not give rise to a neat singular point. For the previous indicators usually some form of scaling is involved in order to identify the transition point, and its value is generally largely influenced by several factors among which the most important is the number of simulated sites, $L$. By contrast, as can be seen in Figs. 1 and 8, the probability of extreme diagonal fluctuations allows to separate the dynamical sides transparently and is valid irrespective of $L$. Furthermore, the resulting singular point shows typical finite-size scalings both in its value and its position, suggesting that it is the precursor of an actual critical point. It is worth to note that experiments in one-dimensional interacting bosons seem also to point to the critical point scenario by studying spatial correlations at long distances after

some time evolution. However, sizes are not large enough to make a sensible extrapolation to the thermodynamic limit [104]. Thus, although our findings seem more compatible with a critical point leading to an actual phase transition, they could signify a change between two distinct, extended regimes. In this sense it is not clear whether both the semi-Poisson and the chaotic regions are finite size effects, disappearing altogether at macroscopic scales and giving rise to an abrupt change from ergodicity to MBL, or if it is a robust characteristic of disordered interacting spin chains that, like the $J_1$-$J_2$ model, undergo a MBL transition. The answer to this question, however, lies out of the scope of this manuscript.

## 6   Conclusion

We have studied the probability of extreme events of the (normalized) fluctuations of the diagonal matrix elements of physical observables around its microcanonical equilibrium value for the $J_1$-$J_2$ disordered quantum spin chain. For intermediate values of disorder this probability exhibits a maximum. Its precise value and the disorder strength at which it is found increase linearly with the system size. We interpret this result as a possible finite-size precursor of the critical point of the ergodic-MBL transition. Below this value of disorder the model is in its chaotic phase, characterized by GOE as in RMT spectra but with long-range deviations due to the Thouless energy. Beyond this value of disorder, an extended region can be identified whose spectral statistics can be described by a family of generalized semi-Poissonian statistics which show level repulsion but not chaotic correlations. Both short and long-range spectral measures can be accurately taken into account by this model. For very large values of disorder, the standard Poissonian statistics associated to the integrability of the localized phase are recovered, where both level repulsion and correlations are lost. Contrary to other ergodicity indicators such as the adjacent level gap ratio or the family of Rényi entropies, this probability is not a monotonous quantity and allows to distinguish these two regimes for any value of the number of sites unambiguously. The main conclusion of our work is that the maximum of the probability of extreme events as represented by the kurtosis excess is an indicator of the hypothetical critical point of the transition. In other words, if the ergodic and MBL phases are indeed connected by an actual phase transition and not by a smooth crossover, *then the critical point must correspond to the value of disorder strength that yields the maximum of the kurtosis excess*. It is interesting to note that the behavior of $\gamma_2(\widetilde{\Delta}_n)$ closely resembles that of a magnetic susceptibility, a robust indicator of a phase transition. This conjecture about the putative position of the critical disorder is complemented with finite-size scaling considerations which are mainly presented in Figs. 3 and 8, where data collapse is shown both for the kurtosis of diagonal matrix elements of the ETH and for the difference between numerical and fitted NNSD histograms, which is a spectral measure. These results therefore provide solid reasons to consider the equality between Thouless energy and Heisenberg energy as a good working criterion to understand the MBL transition in random disordered many-body systems.

We believe our results are an important contribution for a better understanding of the ergodic-MBL transition and the MBL phase itself, opening up several new avenues of research. These ideas immediately call for a study of their relationship with Griffiths effects and their generality in other many-body systems with MBL transition. It should be mentioned that the momentum distribution is a quantity *experimentally accessible* in time-of-flight experiments with cold atoms [105]. By performing many copies of the same experiment with different disorder strengths, our results could be tested experimentally. One of the main questions in the field is the relationship between the standard one-body Anderson localization transition and the Anderson localized phase and the many-body localization transition and phase. Our description of the spectral statistics of the MBL phase with a generalized semi-Poisson model

which is known to describe the critical behavior of the Anderson model at high dimensions should be an important motivation for the current effort of understanding the relationship between the MBL transition and the Anderson localization transition in high dimensional lattices [106]. The main problem in the study of many-body disordered systems, which is also the main limitation of the results presented in this paper, is the scaling to the thermodynamic limit, $L \to \infty$. Without a complete underlying scaling theory for MBL systems, trying to solve the problem by simply expanding the range of system sizes amenable to diagonalization schemes may, however, be insufficient to obtain satisfying solutions to the open questions in the field [46]. However, we believe that our results could serve as guidance for the search of a theoretical framework, probably in the form of a more complete renormalization group approach, capable of overcoming this limitation.

# Acknowledgements

**Author contributions**   A. L. Corps and A. Relaño designed the numerical protocol and performed the calculations. A. L. Corps wrote the first version of the manuscript. All authors discussed the results and contributed to the final version of the manuscript.

**Funding information**   This work has been supported by the Spanish Grant PGC2018-094180-B-I00 (MCIU/AEI/FEDER, EU), CAM/FEDER Project No. S2018/TCS-4342 (QUITEMAD-CM), and CSIC Research Platform on Quantum Technologies PTI-001.

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
