# Peer review of "Signatures of a critical point in the many-body localization transition"

_SciPost Physics, doi:SciPost Phys. 10, 107 (2021)_

## Round 1 · Referee Report · Anonymous · 2021-1-5

Strengths
1-Clearly written
2-Relatively clean data
Weaknesses
1-Unclear what is really new
2-Most of the data only presented for a single system size (L = 16), so no scaling analysis attempted (even if probably not really feasible).
Report
The paper studies the many-body localization transition using, a) the kurtosis excess of the diagonal fluctuations of the momentum distribution, and then b) level statistics beyond nearest levels. The level statistics is found to behave randomly below the maximum of the excess kurtosis obtained at disorder strength $\omega_c$, but with a Thouless energy that decreases as it approaches $\omega_c$ where it roughly becomes equal to the Heisenberg energy. Beyond this $\omega_c$ the distribution of diagonal fluctuations is semi-Poissonic.
In the abstract the authors claim that this "unambiguously identifies a possible finite-size precursor of a critical point." How one unambiguously identifies something that is only possibly something is a bit beyond me, but I think this phrase captures the essence of their results. Their results seem fully consistent with what is already known. The exact same Thouless energy considerations were performed by the same authors in their Ref. [45], focusing n the ergodic side of the transition, albeit in a different model (XXZ instead of J1-J2, I don't think this has any major effect). Semi-Possion level statistic was discussed for example by Serbyn and Moore (in Ref. [24]). The excess kurtosis of a different observable has also been explored, for example by Colmenarez et al ([SciPost Phys. 7, 064 (2019), https://scipost.org/SciPostPhys.7.5.064, see Fig. 4), which is not cited, and indirectly by Luitz in Ref. [40] (see Fig. 8). Of course, these are in different models and looking at different observables, but the phenomenology remains the same.
Because of the above I feel the manuscript does not obviously meet the criteria for SciPost Physics. It may instead meet the criteria of SciPost Physics Core, especially if something can be said about the scaling dependence of the explored quantities. For example, what happens to the semi-Poisson exponents as a function of system size?
Requested changes
1-It was actually not clear to me what observable is being plotted in Fig. 1, as the description in caption and text is not unambiguous. Are these matrix elements of $\hat{n}_q$ for a fixed value of $q$? If yes, what $q$, if not, what is being used. Please clarify.
2-Around Eq. (10) the authors talk about keeping every two eigenvalues. I couldn't understands what this phrase was supposed to mean. From the equations I concluded that they must mean that the average the level spacing of consecutive levels? Please clarify.
3-Below Eq. (10) the authors write that their Erlang probability density reproduced the semi-Poisson for $\eta = 2$, but their Erlang is $\propto \exp(-s)$ while the semi-Poisson is $\propto \exp(-2s)$. Is there a typo or do the mean something different?
4-Below Eq. (13), maybe they can state what the number variance $\Sigma^2(L)$ is, as I didn't find an explanation for those that don't already know.
5-The main first claim in italic in Section 5 seems to be too strong based on the data presented. Do the authors really believe that they can conclude this from their results? Or if it is a conjecture, can the make that more clear?
6-The authors compare with Serbyn and Moore [24] stating that they agree. I understood [24] to say that the semi-Poisson was on the ergodic side, while here it is on the localised side. Can the authors comment on this?
7-Is there any motivation, beyond their construction, for the semi-Poisson statistics? For example, if I look at the recent work of Prakash, Pixley, and Kulkarni (https://arxiv.org/abs/2008.07547), I understand from them that the spectral form factor as given by Poisson statistics is obtained on the localised side. This is not exactly the same quantity as the authors look at, but it's similar. Can the authors comment on this?
Author: Ángel L. Corps on 2021-02-15 [id 1240]
(in reply to Report 1 on 2021-01-05)
Comment 1
In the abstract the authors claim that this "unambiguously identifies a possible finite-size precursor of a critical point." How one unambiguously identifies something that is only possibly something is a bit beyond me, but I think this phrase captures the essence of their results. Their results seem fully consistent with what is already known. The exact same Thouless energy considerations were performed by the same authors in their Ref. [45], focusing n the ergodic side of the transition, albeit in a different model (XXZ instead of J1-J2, I don't think this has any major effect). Semi-Possion level statistic was discussed for example by Serbyn and Moore (in Ref. [24]). The excess kurtosis of a different observable has also been explored, for example by Colmenarez et al ([SciPost Phys. 7, 064 (2019), https://scipost.org/SciPostPhys.7.5.064, see Fig. 4), which is not cited, and indirectly by Luitz in Ref. [40] (see Fig. 8). Of course, these are in different models and looking at different observables, but the phenomenology remains the same.
Because of the above I feel the manuscript does not obviously meet the criteria for SciPost Physics. It may instead meet the criteria of SciPost Physics Core, especially if something can be said about the scaling dependence of the explored quantities. For example, what happens to the semi-Poisson exponents as a function of system size?
Answer 1 Dear Referee,
Thank you for your input and effort reviewing our manuscript. We have rewritten our work (several paragraphs/explanations and section 3.3 are entirely new) to try to clarify why we think that our manuscript contains novel results and interpretations that warrant publication in SciPost Physics. We think that our work is the first one that explicitly proposes a finite-size scaling precursor of a possible critical point in the thermodynamic limit verifying the following properties:
$(i)$ The critical point can be identified as the maximum of a magnitude, like the magnetic susceptibility, $(ii)$ Both the position and the size of the this maximum scale with the system size, $(iii)$ It provides a definition of two different dynamical phases: disorder strength smaller than the critical value identifies the chaotic phase; larger disorder leads to the semi-Poisson phase (in the sense that it is described by the family of intermediate statistics with uncorrelated spacings but level repulsion depending on $\eta$).
Even though the kurtosis excess has been already analyzed in [SciPost Phys. \textbf{7}, 064 (2019)], none of the previous points were discussed. And the same is true for the Thouless energy and semi-Poisson statistics. The aim of this manuscript is neither to show that the Thouless energy plays a prominent role on the ergodic side of the transition, nor that the semi-Poisson statistics can describe what happens close to the MBL phase. Our main novelty is to propose the existence of two different phases, separated by the maximum of the kurtosis excess, a singular point that manifests a typical finite-size scaling: the chaotic phase --characterized by the role played by the anomalous Thouless energy-- and the semi-Poisson phase --characterized by the parameter $\eta$. As far as we know, our work is the first one in which all these features are integrated in a coherent scenario. Furthermore, our results provide strong support to the view that the critical disorder strength is located where the Thouless time roughly equals the Heisenberg time. This idea has been put forward only very recently, but our results suggest that this criterion may go far beyond spectral quantities as signatures of such a critical point (with the latter definition) also show up in matrix elements of physical observables. Of course, the fact that, say, the system exhibits ergodic properties for small values of disorder is well-known; in this regard the novelty in our manuscript lies in the interpretation that we offer, which differs from some previous results placing the critical point, for example, where the level spacing ratios `cross'.
One of the points which you raised was that a finite-size scaling had not been even attempted; this is now done in the new Section 3.3. and also in Fig. 8 of Section 5. We believe these new results to be quite compelling, given further support to our claims. Regarding the semi-Poisson exponents as a function of $L$, we note that this is a tricky calculation because of the scaling of the value of $\omega$ above which this family of distributions works well. So we show in Fig. 8(b) the difference between numerics and the distributions as a function of $\omega-\omega_{c}(L)$.
In all, we honestly hope you think that these changes are appropriate, and that you may now feel that our manuscript meets the criteria for publication in SciPost Physics.
We have addressed your specific remarks and questions below.
Comment 2
It was actually not clear to me what observable is being plotted in Fig. 1, as the description in caption and text is not unambiguous. Are these matrix elements of $n_{q}$ for a fixed value of $q$? If yes, what $q$, if not, what is being used. PLease clarify.
Answer 2 We rely on an ensemble composed of all the values of $q$. The procedure is as follows: we first calculate $n_{q}$ for a fixed value of $q$; then, we find the diagonal matrix element $O_{nn}$ as explained in Sec. 3.1 and Sec. 3.2. Since we analyze the quantity $\Delta_{n}$, we first need to substract the main trend of each diagonal term $O_{nn}$, which is essentially the microcanonical average $\langle \hat{O}\rangle_{\textrm{ME}}$. Then we divide by the standard deviation $\sigma_{\Delta_{n}}$ [Eqs. (4) and (5)]. We follow these steps for each value of $q=0,1,\ldots,L-1$. After that we put them all together, so the number of 'points' per value of disorder is $L\times N$, where $L$ is the chain length and $N$ is the number of levels in the central region (so as to avoid spurious effects coming from the mobility edge). This idea of `mixing' diagonal elements of observables is common in thermalization studies.
Comment 3
Around Eq. (10) the authors talk about keeping every two eigenvalues. I couldn't understands what this phrase was supposed to mean. From the equations I concluded that they must mean that the average the level spacing of consecutive levels? Please clarify.
Answer 3 The family of intermediate statistics that we are using was initially proposed as that corresponding to a short-range plasma model of the Dyson gas type [PRE \textbf{59}, R1315(R) (1999)]. Not much later, it was shown that `semi-Poisson statistics [are] obtained by removing every other number from a random sequence' [PRE \textbf{60}, 449 (1999)]. If that number is $\eta=2$, one finds the semi-Poisson. We have extended that formula to noninteger values of $\eta$ which is very convenient to study the localization-ergodicity transition. As you pointed out, the underlying idea is expressed via the previous Eq. (10). We have however decided to remove such a discussion which can be found in the original references and may distract the reader from the main message.
Comment 4
Below Eq. (10) the authors write that their Erlang probability density reproduced the semi-Poisson for $\eta=2$,but their Erlang is $\propto \exp(-s)$ while the semi-Poisson is $\propto \exp(-2s)$. Is there a typo or do they mean something different?
Answer 4 One needs to calculate the level spacing distribution $P(s)$ corresponding to Eq. (10). While it is true that the sum of $\eta$ unitary exponential random variables has the Erlang distribution with unitary shape, say, $\mathcal{P}(s,\eta)$, one still needs to normalize the distribution by $\eta$ (see Eq. (10)), so that our working distribution is $P(s;\eta)=\eta \mathcal{P}(\eta s;\eta)$ (Eq. (11)). It can be readily seen that $P(s;\eta=2)=4s\exp(-2s)$. As in your previous remark, we have eliminated this discussion because it can be equally well found in the original references and it is not essential to make our point.
Comment 5
Below Eq. (13), maybe they can state what the number variance $\Sigma^{2}(L)$ is, as I didn't find an explanation for those that don't already know.
Answer 5 Thank you for your suggestion. As in your comments 3 and 4 above, we have decided to remove that short discussion on the number variance as we never actually use it and it might deviate the readers' attention.
Comment 6
The main first claim in italic in Section 5 seems to be too strong based on the data presented. Do the authors really believe that they can conclude this from their results? Or if it is a conjecture, can they make that more clear?
Answer 6 Thank you for your remark. The claim at the beginning of Sec. 5 to which you refer is indeed a conjecture based on the previous results that we obtain. We have no way of `proving' it. In any case, we should highlight that this conjecture is compatible and goes in the same direction as other works indicating intriguing behavior at the disorder strength where the Thouless time roughly equals the Heisenberg time, see, e.g., PRE \textbf{102}, 062144 (2020), PRB \textbf{102}, 064207 (2020), PRL \textbf{115}, 046603 (2015), PRB \textbf{97}, 201105 (2018), PRB \textbf{96}, 104201 (2017). We have clarified that this is a conjecture in the main text, as required.
Comment 7
The authors compare with Serbyn and Moore [24] stating that they agree. I understood [24] to say that the semi-Poisson was on the ergodic side, while here it is on the localised side. Can the authors comment on this?
Answer 7 We have gone through [PRB \bf{93}, 041424(R) (2016)] again. As far as we understand, our results seem fully compatible with theirs. They argued that [the flow from Wigner-Dyson to Poisson statistics is a two-stage process] and that [at the second stage, the gas of eigenvalues has local interactions and the level statistics belongs to a semi-Poisson universality class.] This is what we observe as well. At the second stage of the transition, once random matrix correlations have been lost between levels separated by any distance, the model of intermediate statistics begins providing a good description, although not perfect; more complex models have been devised as well, but our intention here is not to accurately describe the level statistics, but to show that they seem compatible with short range local interactions of the type of Eq. (11). We do not claim to provide a perfect description; our goal in this part is to show that when the maximum of the kurtosis of $\Delta_{n}$ is reached, there is an abrupt change in the kind of interactions which is reflected in the level statistics. It is interesting to note that our findings agree with this picture not only qualitatively but quantitatively; they conclude that `there are two steps of the spectral statistics flow, one with long-range interactions (the plasma model) and one with local interactions, and the boundary between the two is found numerically to coincide with the onset of a Griffiths phase '. They find exact semi-Poisson statistics ($\eta=2$) at the value of $\omega$ where the Griffiths phase is often identified, but also argue that finite-size effects might play an important role in a precise identification. We could not find good agreement with the family of intermediate statistics $P(s;\eta)$ before the strength $\omega_{c}(L)$ at which $\gamma_{2}(\widetilde{\Delta}_{n})$ is maximum (where Griffiths effects are usually identified). This family seems to provide a good description at the second stage of the transition. In any case, since our $\eta:=\beta+1$, our results seem to be in agreement with theirs.
Comment 8
Is there any motivation, beyond their construction, for the semi-Poisson statistics? For example, if I look at the recent work of Prakash, Pixley, and Kulkarni (https://arxiv.org/abs/2008.07547), I understand from them that the spectral form factor as given by Poisson statistics is obtained on the localised side. This is not exactly the same quantity as the authors look at, but it's similar. Can the authors comment on this?
Answer 8 There are several reasons for using the family of intermediate statistics Eq. (11) (including the semi-Poisson). One of them is its clear mathematical tractability. However, physically this is not the actual, relevant reason. The fact is that, as we write, in the one-body metal-insulator transition (i.e., Anderson localization) this generalized semi-Poisson model seems to describe the spectral statistics of the critical region with a value of $\eta$ that changes from 2 to 1 as the dimensionality is increased as has been numerically investigated up to six spatial dimensions [e.g., PRB \textbf{75}, 174203 (2007); ideas in the same direction have been also presented elsewhere]. Understanding the similarities (and divergences) between the one-body localization and the many-body localization phenomena is a fascinating endeavor that has been undertaken during the last decade. Other authors have formulated highly involved plasma models inspired by these ideas ([41,42], for instance). Our intuition was to follow the same direction. However, as we have mentioned before, we do not intend to provide an extremely accurate description of the level statistics in the transition, but to show that the maximum of $\gamma_{2}(\widetilde{\Delta}_{n})$ roughly occurs at the value of disorder beyond which chaotic correlations are completely lost and levels start interacting repulsively only locally; this is a major conceptual difference.
In the preprint arXiv:2008.07547 the authors give analytic results for the spectral form factor deep in the localized phase (i.e., when level statistics are described by Poisson statistics), and discuss deviations from this result due to various effects. The main difference with the family of intermediate statistics (generalized semi-Poisson) of our work is that their result is expected to be valid if $\omega\gg \omega_{c}(L)$, whereas ours is proposed as valid for all $\omega>\omega_{c}(L)$. Hence, their result is compatible with ours when $\omega\gg \omega_{c}(L)$ (i.e., $\eta\to 1$).
Author: Ángel L. Corps on 2021-02-15 [id 1239]
(in reply to Report 2 on 2021-01-25)Comment 1
Answer 1 Dear Referee,
We thank you for your input and effort reviewing our manuscript. In particular we are grateful for your view that our work may help introduce a fresh idea in the context MBL transitions. We have followed your advice (see summary of changes, `major changes') and new results and interpretations are given to highlight the consistency (and inconsistency) with other ways of thinking about the MBL transitions. It is our hope that these changes have been done to your satisfaction.
Comment 2
Answer 2 This word has been removed from the abstract accordingly. Also, the abstract has been almost entirely rewritten to try and express more clearly the novelty of our work.
Comment 3
Answer 3 We agree with you and the other referee that this reference was missing. This has been remedied.
Comment 4
Answer 4 Thank you for this remark. In Fig. 6 (now Fig. 7) the color is only used to identify the values of the disorder $\omega$ which are also considered in previous Figs. 3, 4 and 5 (now Figs. 4, 5 and 6) (though not any other figure). It is only intended as a visual guide. This clarification has been added to the caption of the figure.

---

## Round 1 · Referee Report · Anonymous · 2021-1-25

Strengths
The manuscipt has the potential to present a relatively new and fresh idea in the context mbl transitions
Weaknesses
It is far from obvious to readers what is the new contribution of this work
Report
The manuscript by Corps et al investigates the many-body localization transition by studying the distributions of observable matrix elements and the spectral statistics. For the first, they study the probability of extreme events by means of the kurtosis excess, and for the second, they study the power spectrum of the \delta_n spectral statistics.
Both quantities, the kurtosis, and the power spectrum, have to some extent already been studied in the literature. The power spectrum was studied by the same authors in the XXZ model in Ref. 45, and the description within the semi-poisson statistics recently in Ref. 90. The kurtosis of local observables in the XXZ model was studied by Colmenarez et al, SciPost Phys. 7, 064 (2019). However, the latter work [that followed previous work in Ref.40] interpreted the analysis of the observable matrix elements from the "standard" numerical perspective of the mbl transition - which takes place where the r level spacing statistics approaches the poisson value (this is w ~ 3.7 in the XXZ model, and probably more than twice larger for the J1J2 model, I suppose w >~ 8). The manuscript by Corps et al takes a different perspective, and they claim, according to their abstract, that the transition takes place when the Thouless energy becomes equal to the Heisenberg energy. This transition point does not quantitatively agree with the one that is conventionally obtained from the r statistics. This result may seem surprising, and the authors provide some solid evidence to support their claim.
I therefore see the potential of the manuscript to present a relatively new and fresh idea in the context mbl transitions. In this sense, I may recommend the manuscript for publication in SciPost Phys. That said, I think it is at the moment far from obvious to readers what is the new contribution of this work, and in which sense does it connect to some previous works or it proposes a new interpretation. Hence I encourage the authors to rewrite their manuscript, and to make clear statements about its novelty and (in)consistency with other proposals for the mbl transition.
Requested changes
- As suggested by the first referee, the authors should remove the word “unambiguous” from the abstract.
- They should cite Colmenarez et al, SciPost Phys. 7, 064 (2019), which also studied the kurtosis excess.
- They should explain the meaning of colors of filled symbols in Fig.6

---

## Round 2 · Referee Report · Anonymous · 2021-3-18

Report
The authors have improved their manuscript and clarified in their text and response what they consider to be the main advances of their work. They have also clarified that strong statements made in the first version are conjectures inspired by their data rather than definite conclusions.
There are still a couple of points in their response and updated manuscript that I am a bit unsure about. First, there is no sense, as far as I can see, that $\omega_c(L)$, obtained as the maximum of the kurtosis excess, is a $\textit{singular point}$. It is just a feature of finite size data and there is nothing singular about it. In particular, since it clearly and strongly flows with system size, the authors at one point write that they can not strictly conclude if it is reflecting an underlying critical point or just a finite-size crossover, in which it would not even be singular in the thermodynamic limit. In my mind it is confusing to talk about $\omega_c(L)$ as a singular point when what it really is is just a way of estimating the critical disorder strength from finite size data. It may be a nice way of doing this, and it does correlate well with estimates of when the Thouless energy becomes equal to the Heisenberg energy, but I do not see that it has otherwise a higher status than some of the other ways of estimating the critical disorder strength in finite size systems.
It is true that this way is better than using crossings in level spacing statistics (or r ratios), but in my mind it is well understood that these crossing are not very accurate. In their response to my first report the authors claim that the kurtosis excess is special since it gives a peak at (or close) to the transition. On page 20 in their manuscript they also compare this with entropies stating that they change monotonically with disorder strength, citing several references including Ref. [37] (Kjäll et al). This is not fully accurate since for example in Ref. [37] the fluctuations of the entanglement entropy are shown to peak at the transition and the maximum of the variance could be used in the very same way as here to estimate the critical disorder strength (Ref. [37] instead used finite-size scaling collapse to estimate the critical disorder strength). So the kurtosis excess is not the first or only quantity to have the features the authors claim is novel. I think this is OK. It doesn't really affect the potential value of this work but I think it is not useful to write and discuss it in a way where it is made to seem like it reveals more than it actually does.
The above are comments on the presentation and the discussion and interpretation of the data. I think it is exaggerating a bit what can be read from the data, and how it positively compares with other data. I don't think this is needed (or useful) as some of the data is actually quite nice. I like figures 5 and 6 for example, and the way of analysing this and the physics that comes from it, is a useful addition to the literature. I think it is consistent with what is already known, and even if there isn't really anything fundamentally new the work provides a nice way of analysing data.
One last comment: in the authors reply they state that the concept of Thouless energy becoming the Heisenberg energy has only recently entered the discussion. This is not strictly true. Arguably this goes back to Thouless' early work, albeit in the Anderson transition. But even in the context of MBL it goes back further, and it was for example discussed by Serbyn, Papić and Abanin in Phys. Rev. B 96, 104201 (2017), which is not cited.
In any case, if the authors tidy up these last presentation issues (see requested changes ) I think it deserves publication. I am undecided if I think it belongs more to the SciPost Physics of SciPost Physics Core, so I'll live it up to the editorial college to decide in case the editor-in-charge goes forward with publication. If it does pass for SciPost Physics, it would maybe be in expectation 3. (Open a new pathway in an existing or a new research direction, with clear potential for multipronged follow-up work;), but for SciPost Physics Core it satisfies 2. (Detail one or more new research results significantly advancing current knowledge and understanding of the field.)
Requested changes
1-Clarify in what sense $\omega_c(L)$ is a singular point or, more likely, if it is not a singular point, rephrase the discussion of it as being a finite-size estimator of the critical value for the disorder strength.
2-If the maximum of the kurtosis excess $\omega_c(L)$ is fundamentally different signature as a precursor to the MBL transition than, say the maximum in the variance of the eigenstate entanglement entropy, clarify this difference and contrast with quantities that behave in a similar way (instead of only contrasting it with quantities that do not). Note that I used the word fundamentally different since of course it is different in the details of the quantities that are being looked at.

---

## Round 2 · Referee Report · Anonymous · 2021-3-30

Report
The authors have made a considerable effort to rewrite the manuscript, to improve the presentation, and to address the criticism from the referees raised in the first round. In my opinion the authors did a good job and have sharpened the message of their analysis.
The main achievements of their work are given below:
- They perform a thorough analysis of both the kurtosis excess of the observable matrix elements and the long-range spectral statistics via the averaged power spectrum.
- They establish the connection between the peak of the kurtosis excess and the breakdown of the RMT description, and identify this point as a possible candidate for the transition point in the thermodynamic limit.
- These results, combined together, constitute a new result in the field of widely studied disordered spin-1/2 chains.
For these reasons, I recommend the manuscript to be published in SciPost Phys as is.

---

## Round 2 · Author Response

Thank you for sending us the Referees' comments. We have revised our manuscript accordingly. Below you can find the response to each of the Referees.
Yours sincerely,
The Authors

---

## Round 2 · List of Changes

We have integrated the comments/suggestions of the referees with our work. Here we list them.
(A) Major changes
A1. The abstract has been rewritten to clarify the novelty of our manuscript (in particular the word `unambiguous' has been removed).
A2. Added Section 3.3., where a finite-size scaling of the kurtosis excess $\gamma_{2}(\overline{\Delta}_{n})$ is presented. Added Fig. 3 consisting of three panels.
A3. Previous Fig. 7 has been modified and the new Fig. 8 is now located in Section 5. This figure now consists of two panels, and it contains finite size-scaling considerations for both thermalization and spectral statistics, which offers an integrated scenario. \\
(B) Other changes
B1. Added several references.
B2. In general, some parts of the text involving well-known previous results have been either simplified or entirely removed (the reader is thus simply referred to original references) as they could have influenced the perception of novelty.
B3. The meaning of colored points in previous Fig. 6 (now Fig. 7) has been clarified.
B4. We have stated more explicitly that the first claim in Section 5 is a conjecture.
B5. Several clarifications and explanations have been added throughout.

---

## Editorial Decision

published